# Identification and characterization of intermediate states in mammalian neural crest cell epithelial to mesenchymal transition and delamination

**Ruonan Zhao[1,2], Emma L Moore[1], Madelaine M Gogol[1], Jay R Unruh[1], Zulin Yu[1], Allison R Scott[1], Yan Wang[1], Naresh K Rajendran[1], Paul A Trainor[1,2]***

[1]Stowers Institute for Medical Research, Kansas City, United States; [2]Department of Anatomy and Cell Biology, University of Kansas Medical Center, Kansas City, United States

**\*For correspondence:**
pat@stowers.org

**Abstract** Epithelial to mesenchymal transition (EMT) is a cellular process that converts epithelial cells to mesenchymal cells with migratory potential in developmental and pathological processes. Although originally considered a binary event, EMT in cancer progression involves intermediate states between a fully epithelial and a fully mesenchymal phenotype, which are characterized by distinct combinations of epithelial and mesenchymal markers. This phenomenon has been termed epithelial to mesenchymal plasticity (EMP), however, the intermediate states remain poorly described and it's unclear whether they exist during developmental EMT. Neural crest cells (NCC) are an embryonic progenitor cell population that gives rise to numerous cell types and tissues in vertebrates, and their formation and delamination is a classic example of developmental EMT. However, whether intermediate states also exist during NCC EMT and delamination remains unknown. Through single-cell RNA sequencing of mouse embryos, we identified intermediate NCC states based on their transcriptional signature and then spatially defined their locations in situ in the dorsolateral neuroepithelium. Our results illustrate the importance of cell cycle regulation and functional role for the intermediate stage marker *Dlc1* in facilitating mammalian cranial NCC delamination and may provide new insights into mechanisms regulating pathological EMP.

## eLife assessment

This **fundamental** study reports **compelling** findings that intermediate states exist in epithelial-mesenchymal transition (EMT) during natural development and differentiation of mammalian neural crest cells, similar to recent reports in cancer. The authors determined that there were at least two paths to delamination and migration - one that occurs during S-phase of cell cycle and another during G2/M phase, and that the process of delamination is not restricted to cell fate. Finally, the authors showed that expression of Dlc1 may be used to identify cells in an intermediate state of EMT as well as their spatial location in the mouse embryo. The work will be of interest to developmental biologists, neurobiologists and cancer researchers.

## Introduction

Epithelial to mesenchymal transition (EMT) is a cellular process that converts epithelial cells to mesenchymal cells with migratory potential (*Hay, 1995*). EMT plays an essential role in various developmental and pathological processes such as embryonic morphogenesis, wound healing, tissue fibrosis,

and cancer progression (*Zhao and Trainor, 2023*). Studies of EMT, particularly in the field of cancer biology, have increased exponentially in the past 5 years due to the implied role of EMT in numerous aspects of malignancy such as cancer cell invasion, survival, stemness, metastasis, therapeutic resistance, and tumor heterogeneity (*Yang et al., 2020a*).

EMT has traditionally been considered a binary process comprising either full epithelial cell or mesenchymal cell states. However, studies of cancer have uncovered an alternative scenario termed epithelial to mesenchymal plasticity (EMP), in which multiple intermediate states exist along the EMT spectrum (*Dong et al., 2018*; *Gonzalez et al., 2018*; *Huang et al., 2013*; *Karacosta et al., 2019*; *Kumar et al., 2019*; *Pastushenko et al., 2018*). More specifically, in some cases of head and neck cancer, primary tumors and matching lymph nodes contain a subpopulation of tumor cells in a partial EMT state, as defined by their expression of both epithelial marker genes and mesenchymal marker genes (*Puram et al., 2017*). Moreover, cells in this partial EMT state are highly invasive and located at the leading edge of tumors in vivo. Similarly, in mouse xenograft models of skin cancer and breast cancer, the absence of epithelial cellular adhesion molecules in several cell populations was discovered to represent early EMT hybrid states as they also expressed both Vimentin and Cytokeratin 14 at intermediate levels and consequently had high metastatic potential (*Pastushenko et al., 2018*). Partial EMT states have also been recognized in lung cancer and ovarian cancer (*Gonzalez et al., 2018*; *Karacosta et al., 2019*), with tumor cells expressing both the epithelial marker E-cadherin and the mesenchymal marker Vimentin. Besides primary tumors, EMT intermediate states have also been identified in circulating tumor cells from patient samples (*Yu et al., 2013*). Even though EMT intermediate states have been discovered in numerous studies, few have focused on describing and understanding the molecular and cellular mechanisms governing or defining each intermediate state due to the challenges of studying cancer initiation and progression in vivo.

Since EMT during embryogenesis and cancer progression have been shown to share analogous phenotypic changes that involve similar core transcription factors and molecular mechanisms, it was proposed that the initiation and development of carcinoma could be attributed to an unusual activation of EMT factors involved in normal developmental processes (*Hay, 1995*). However, compared to tumorigenesis, it remains largely unknown whether intermediate or transition states exist or play a role in classic developmental EMT. Therefore, identifying and characterizing intermediate states during developmental EMT can further our understanding of the cellular processes, and molecular signaling networks that regulate EMP.

NCC formation is a classic example of developmental EMT (*Lee et al., 2013*; *Zhao and Trainor, 2023*). NCC are a migratory progenitor cell population unique to vertebrates. Formed during neurulation in the dorsolateral domain of the neural plate, EMT facilitates their delamination from the neuroepithelium and migration throughout the body, where they differentiate into neurons and glia of the peripheral nervous system, pigment cells in the skin, craniofacial bone and cartilage, as well as many other cell types (*Bhatt et al., 2013*; *Dash and Trainor, 2020*; *Lièvre and Douarin, 1975*; *Trainor, 2005*; *Weston, 1983*). Disruption of NCC delamination and migration can result in developmental abnormalities, referred to as neurocristopathies (*Achilleos and Trainor, 2015*; *Watt and Trainor, 2014*), hence it is important to study the mechanisms that regulate mammalian NCC development.

We performed single-cell RNA sequencing (scRNA-seq) to identify and define intermediate transcriptional and cellular states during mouse cranial NCC EMT and delamination. We identified two NCC EMT intermediate populations distinguished by their S or G2/M cell cycle phase state during delamination. Interestingly, trajectory analyses reveal that these distinct intermediate populations are formed simultaneously, and independently, but then converge into a single or common pool, suggesting they do not have distinct fates following migration. This is consistent with the known plasticity and potency of early migrating NCC (*Golding et al., 2000*; *Sandell and Trainor, 2006*; *Trainor and Krumlauf, 2000a*; *Trainor and Krumlauf, 2000b*; *Trainor and Krumlauf, 2001*). Transcriptional profiling revealed that the intermediate NCC populations could also be defined by unique transcriptional signatures, including differential expression of genes involved in cell protrusion, such as *Dlc1*, *Pak3*, and *Sp5*. Further interrogation using signal amplification by exchange reaction for multiplexed fluorescent in situ hybridization (SABER-FISH) revealed that these intermediate NCC populations were spatially localized in the dorsolateral region of the neural plate. In addition, knocking down the NCC EMT intermediate marker *Dlc1* led to a significant reduction in the number of migratory NCC, which revealed a critical role for *Dlc1* in the regulation of mouse cranial NCC delamination. Overall, our

findings provide novel, detailed, high-resolution descriptions of the intermediate cell populations, and transcriptional states that occur during cranial NCC EMT and delamination in mouse embryos. Our work further illustrates that molecular characterization of NCC EMT intermediate states can reveal essential regulatory components of mouse NCC EMT and delamination. These results shed light on similar mechanisms of NCC EMT and delamination in other mammalian species and will also serve as a resource for the community. In addition to NCC EMT, our work may also help to inform the phenotypic changes and corresponding gene regulatory control of EMP in other developmental EMT events as well as pathological conditions such as tissue fibrosis and cancer progression.

## Results

### Identification of intermediate stages during mouse cranial NCC EMT and delamination

To investigate the biological process and mechanisms governing mouse cranial NCC EMT and delamination, we performed scRNA-seq on dissociated cranial tissues isolated from E8.5 mouse embryos with 7–9 somites (*Figure 1A*). More specifically, embryos were collected from two transgenic mouse lines: Wnt1-Cre;Rosa^LSL-eYFP (*Chai et al., 2000*) and Mef2c-F10N-LacZ (*Aoto et al., 2015*; *Figure 1A*). In E8.5 Wnt1-Cre;Rosa^LSL-eYFP embryos, YFP is expressed by *Wnt1+* neuroepithelial cells located in the dorsolateral neural plate, which encompasses premigratory NCC (*Figure 1B*). Consequently, Wnt1-Cre;Rosa^LSL-eYFP labels premigratory and migratory NCC and other lineage labeled cells derived from the *Wnt1+* cell population (*Figure 1B*; *Chai et al., 2000*). In contrast, Mef2c-F10N-LacZ predominantly labels migratory NCC, with LacZ activity driven by the F10N enhancer of the *Mef2c* gene (*Figure 1B*; *Aoto et al., 2015*). The two different transgenic lines allowed us to distinguish premigratory from migratory NCC spatially within an embryo, but also later bioinformatically following single-cell dissociation and RNA-seq.

The scRNA-seq data was processed and analyzed as previously described (*Falcon et al., 2022*). We initially identified six major cell or tissue types present in E8.5 mouse embryonic cranial tissues based on the differential expression of classic cell or tissue type-specific markers, and we clustered the data accordingly (*Figure 1C and D*; *Figure 1—figure supplement 1*). For example, *Sox1* and *Sox2* were used to delineate neural ectoderm, whereas *Cdh1* (E-cadherin) was used to define non-neural ectoderm. *eYFP*, *LacZ*, *Sox10,* and *Twist* delineated migrating NCC. *Tbx1* was primarily used as a marker of mesoderm cells and *Kdr* (*Vegfr2*) demarcated mesoderm-derived endothelial cells. We then bioinformatically segregated the cranial NCC cluster, which includes both premigratory and migratory NCC, and divided it into five subclusters at a resolution of 0.26 (*Figure 2A*).

To characterize these NCC subclusters, we then interrogated the expression of known neuroepithelial and neural plate border markers, as well as genes expressed by NCC during their specification and migration (*Supplementary file 1*; *Echelard et al., 1994*; *Hafemeister and Satija, 2019*; *Lee et al., 2013*; *Murdoch et al., 2012*; *Parr et al., 1993*; *Sauka-Spengler and Bronner-Fraser, 2008*; *Wood and Episkopou, 1999*). A high percentage of cells in subclusters 0–3 express elevated levels of markers of migratory NCC (*Vim, Sox10, Twist1*), while only subclusters 0 and 1 exhibit high levels of expression of NCC specifier genes (*Zeb2, Pax3, Nr6a1, Sox9, Foxd3, Snai1*) (*Figure 2B*; *Cheung et al., 2005*; *Dottori et al., 2001*; *Hari et al., 2012*; *Kobayashi et al., 2020*; *Lee et al., 2013*; *Li et al., 2000*; *Murdoch et al., 2012*; *Schorle et al., 1996*; *Soo et al., 2002*; *Van de Putte et al., 2003*). Since NCC specifiers are downregulated as NCC migrate and later differentiate into specific lineages, the combinatorial expression of genes suggests that subclusters 0 and 1 likely represent an earlier stage of NCC delamination and migration than subclusters 2 and 3. This conclusion was further verified by integrating previously published networks of genes that represent an early migratory NCC program versus a late migratory NCC program (*Soldatov et al., 2019*). Early migratory NCC program genes are expressed by the majority of migratory NCC whereas late migratory NCC program genes are only expressed in a subset of migratory NCC as they have already begun to mature.

Subclusters 0–3 each exhibit significant expression of early migratory NCC program genes (*Figure 2—figure supplement 1*). Furthermore, subclusters 2 and 3 express a significantly higher level of late migratory NCC program genes than subclusters 0 and 1, which demonstrates that subclusters 0 and 1 contain early migratory NCC, whereas subclusters 2 and 3 comprise late migratory NCC (*Figure 2—figure supplement 1*). Interestingly, we also observed that a small percentage of

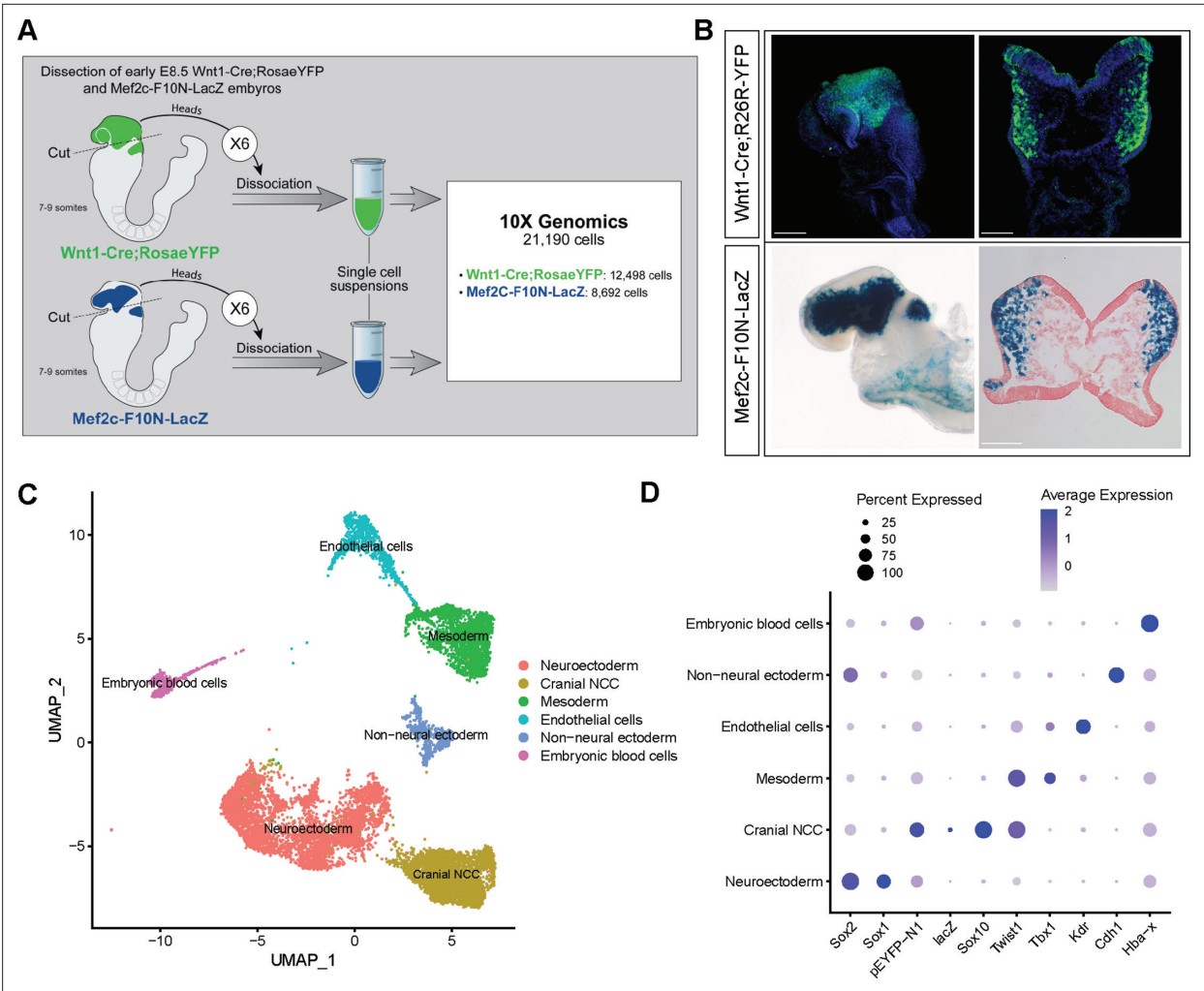

**Figure 1.** Single-cell RNA sequencing analysis of mouse early E8.5 cranial tissues. (**A**) Schematic of experimental design. Wnt1-Cre;Rosa[LSL-eYFP] and Mef2c-F10N-LacZ embryos with between 7 and 9 somites (6 each) were dissected and cranial tissues anterior to rhombomere 3 were collected. Tissues were dissociated into single-cell suspensions before being processed through the 10x Genomics pipeline. The final dataset used for analysis consisted of 21,190 cells (12,498 cells from Wnt1-Cre;Rosa[LSL-eYFP] and 8692 from Mef2c-F10N-LacZ) and 29,041 genes. (**B**) YFP and LacZ staining of E8.5 Wnt1-Cre;Rosa[LSL-eYFP] and Mef2c-F10N-LacZ embryos and 10 μm cranial transverse sections. YFP (green) labels cells located in the dorsal neuroepithelium and their lineages. As a result, both premigratory and migratory neural crest cells (NCC) are marked by YFP expression. LacZ (blue) labels migratory NCC. (**C**) Uniform Manifold Approximation and Projection (UMAP) and clustering of six major tissue types in the cranial region of E8.5 mouse embryos: cranial NCC, neuroectoderm, non-neural ectoderm, mesoderm, endothelial cells, and embryonic blood cells. (**D**) Dotplot showing the expression of tissue-specific markers used for cluster identification. Dot size indicates the percentage of cells in each corresponding cluster (y-axis) that expresses a specific gene (x-axis). Dot color intensity indicates the average expression level of a specific gene in a cell cluster. Scale bars: whole embryos 200 μm; embryo sections 100 μm.

The online version of this article includes the following figure supplement(s) for figure 1:

**Figure supplement 1.** Expression of tissue-specific marker genes that identify six major cell type clusters in early E8.5 mouse embryonic cranial tissues.

subcluster 0 cells express early migratory NCC genes, but at a lower intensity compared to subcluster 1. This implies that subcluster 0 might also contain premigratory NCC that do not yet express any migratory NCC genes (*Figure 2—figure supplement 1*). Despite similar expression profiles, NCC subclusters 2 and 3 possibly represent undifferentiated NCC-derived mesenchyme tissue in different parts of the head. More specifically, subcluster 2 displays a high level of expression of pharyngeal arch NCC mesenchyme marker *Dlx2* (*Figure 2—figure supplement 1*; *Bulfone et al., 1993*). In contrast, subcluster 3 expresses a frontonasal mesenchyme marker *Alx1* (*Figure 2—figure supplement 1*; *Iyyanar et al., 2022*). Consistent with these observations, neither population expresses a high level of

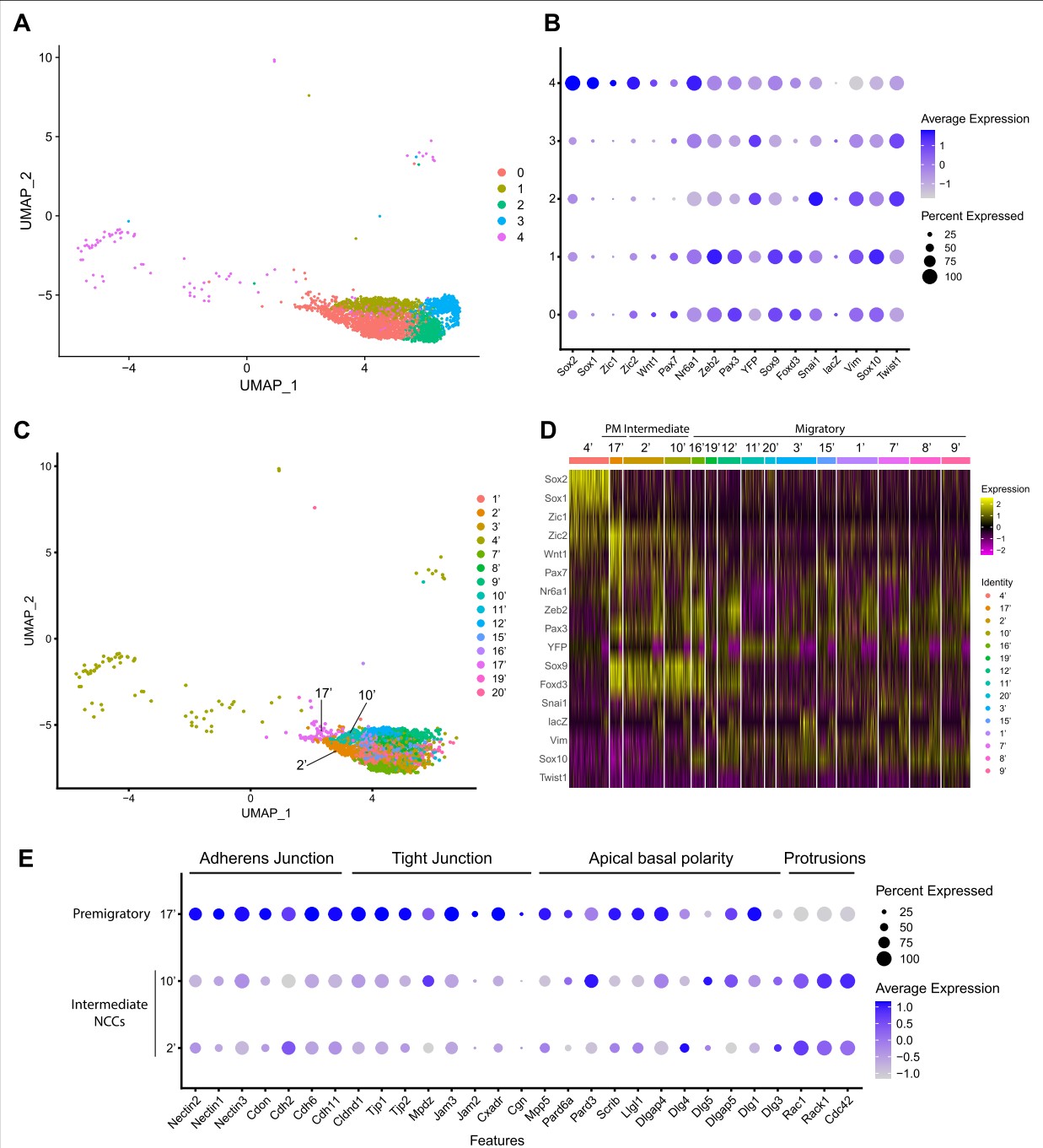

**Figure 2.** Expression of neural crest cell (NCC) development-related genes and epithelial to mesenchymal transition (EMT) functional genes identifies NCC EMT intermediate populations. (**A**) Uniform Manifold Approximation and Projection (UMAP) and re-clustering of the cranial NCC cluster into five smaller subclusters at a resolution of 0.26. (**B**) Dotplot showing the expression of NCC development-related genes in five cranial NCC subclusters. (**C**) UMAP and re-clustering of the early migratory NCC subclusters 0, 1, and 4 into smaller subclusters at a resolution of 2.0. (**D**) Heatmap showing the expression of NCC development-related genes in the smaller early migratory NCC subclusters at a resolution of 2.0 shown in (**C**). High levels of expression are indicated in yellow, and low levels of expression are indicated in pink. Based on the gene expression profile of each subcluster, subcluster 17′ was determined to be premigratory NCC; subcluster 2′ and 10′ are EMT intermediate NCC; the remaining subclusters are migratory NCC. (**E**) Dotplot showing the expression of EMT functional genes in premigratory NCC subcluster 17′ and intermediate NCC subclusters 2′ and 10′. EMT intermediate NCC display reduced expression of adherens junction, tight junction, and apical basal polarity genes compared to premigratory NCC, whereas protrusion-related genes are upregulated in intermediate NCC.

The online version of this article includes the following figure supplement(s) for figure 2:

**Figure supplement 1.** Additional gene expressions that support the identity of cranial neural crest cell (NCC) subclusters at 0.26 and 2.0 resolutions.

neurogenic lineage markers such as *Nrp1* and *Nrp2*, which is indicative of their undifferentiated state (*Figure 2—figure supplement 1*; *Lumb et al., 2014*).

Subcluster 4 exhibits a unique transcriptional profile distinct from subclusters 0–3 (*Figure 2B*). Higher expression of *Sox1* and *Sox2* is indicative of a neuroepithelial identity for subcluster 4 (*Figure 2B*). During NCC formation, *Sox1* and *Sox2* are downregulated in the dorsal neural plate border domain but remain strongly expressed more ventrally throughout the neural plate (*Supplementary file 1*). Concomitantly, Sox9 and Sox10 are activated in what is known as the SoxB (1/2) to SoxE (9/10) switch (*Mandalos et al., 2014*; *Remboutsika et al., 2011*; *Wakamatsu et al., 2004*). Consistent with this model, overexpressing *Sox2* in the dorsal neural tube has been shown to repress NCC specification, whereas overexpressing *Sox9* and *Sox10* have been shown to precociously promote NCC formation (*Aybar et al., 2003*; *Cheung and Briscoe, 2003*; *Mandalos et al., 2014*; *McKeown et al., 2005*; *Remboutsika et al., 2011*).

To determine if mouse cranial NCC EMT is non-binary and occurs through intermediate or transition states, we further subdivided the five cranial NCC clusters into smaller subclusters (resolution = 2.0; subcluster 1′–21′) and extracted 15 subclusters out of the original early migratory NCC subclusters 0, 1, and 4 (*Figure 2C*; *Figure 2—figure supplement 1*). Through heatmap analysis, expression of the same NCC marker genes as described above was examined within these new subclusters, and the order of the subclusters was arranged according to their combinatorial expression patterns (*Figure 2D*). For example, *Wnt1* and *Sox10* were used to identify NCC transitioning from premigratory to migratory states since *Wnt1* is only expressed in premigratory NCC and is immediately downregulated as NCC delaminate and start to migrate. In contrast, *Sox10* is activated only after NCC have delaminated and begun to migrate. The heatmap shows that a significant number of cells in subcluster 17′ express a much higher level of *Wnt1* than any other subcluster, suggesting that subcluster 17′ comprises premigratory NCC (*Figure 2D*). Consistent with this observation, subcluster 17′ cells also express other neural plate border and NCC specifier genes such as *Zic2*, *Pax7*, *Nr6a1*, *Pax3*, *Sox9*, and *Foxd3,* but does not express migratory NCC markers such as *Sox10* and *Vim* (*Figure 2D*). Subclusters 2′ and 10′ share a similar expression profile to subcluster 17′. However, subclusters 2′ and 10′ express less *Wnt1* and less neural plate border specifiers such as *Zic2* and *Pax7* than subcluster 17′ (*Figure 2D*; *Figure 2—figure supplement 1*). This data suggests that subclusters 2′ and 10′ could represent EMT intermediate states as premigratory NCC transition to migratory NCC during delamination. Lastly, the remaining early NCC subclusters express *Sox10* and *Vim* indicating that they comprise or represent migratory NCC (*Figure 2D*).

To further validate the identity of subclusters 2′ and 10′ as representing intermediate cellular stages of EMT, we assessed the expression of genes associated with adherens junctions, tight junctions, and apical-basal polarity, which are required to maintain epithelial integrity, and cytoskeleton rearrangement that is typically associated with EMT (*Dongre and Weinberg, 2019*; *Matsuuchi and Naus, 2013*; *Radisky and Radisky, 2010*; *Zhao and Trainor, 2023*). Intermediate NCC exhibit a decrease in *Nectin* (e.g. *Nectin, 1, Nectin2, Nectin3*), *Cadherin* (e.g. *Cadh2, Cadh6, Cadh11*), and *Tight Junction Protein* (e.g. *Cldna1, TJp1, TJP2*) gene expression consistent with intercellular tight junction breakdown and degradation of apicobasal polarity (*Figure 2E*). Although many cell junction-related genes (e.g. *Cdh1*) were not expressed by premigratory and intermediate NCC in our data, *Myh9* and *Myh10* were expressed, but didn't exhibit a significant difference or change in their expression levels. This suggests that myosin-II microfilaments localized to the adherens junction-associated circumferential actin belt may not have been impacted at this stage of NCC EMT. At the same time, subcluster 2 and 10 cells exhibit an increase in *Rac*, *Rack1*, and *Cdc42* gene expression which is indicative of cytoskeletal rearrangement and the formation of cell protrusions (*Figure 2E*). These alterations in gene expression are molecular indicators of the cellular mechanisms that underpin EMT (*Figure 2E*).

## Mouse cranial NCC undergo EMT and delamination in S or G2/M phase cell cycle

Since intermediate NCC stages represent a transitional or intermediate phase between premigratory and migratory NCC, the molecular and signaling pathway signatures uniquely expressed by these intermediate NCC can reveal essential regulatory mechanisms governing NCC EMT and delamination. Analysis of cell cycle gene expression, for example, indicated that EMT intermediate NCC populations exhibit distinct cell cycle phase properties or characteristics. NCC in subcluster 2′ primarily

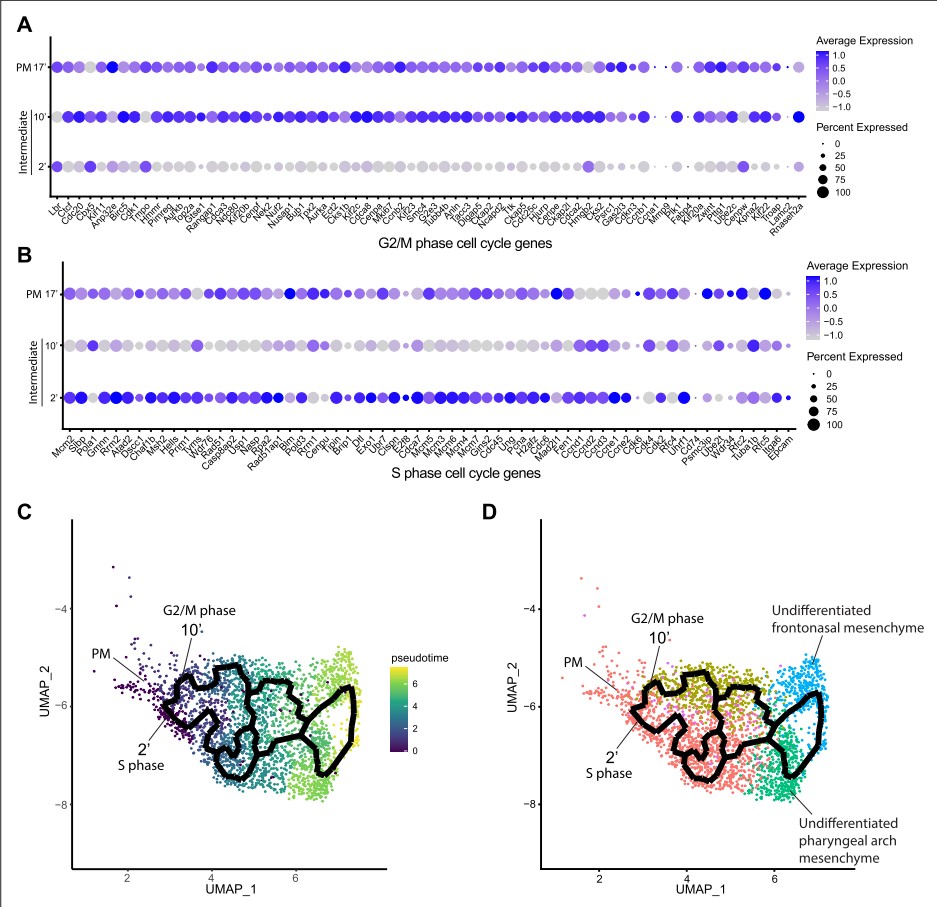

**Figure 3.** Mouse cranial neural crest cell (NCC) delaminate in S phase or G2/M phase cell cycle independently.
(**A**) Dotplot showing the expression of G2/M phase cell cycle genes in premigratory NCC (PM) and epithelial
to mesenchymal transition (EMT) intermediate NCC. G2/M phase cell cycle genes are expressed in PM and
intermediate subcluster 10' cells. (**B**) Dotplot showing the expression of S phase cell cycle genes in PM and
EMT intermediate NCC. S phase cell cycle genes are expressed in PM and intermediate subcluster 2' cells.
(**C**) Pseudotime analysis of the cranial NCC cluster reveals the temporal relationship between intermediate
NCC subclusters 2' and 10'. Dark color indicates early NCC development, and light color indicates later NCC
development. PM and intermediate NCC subclusters represent the earliest developmental timepoints among
all cranial NCC. (**D**) Trajectory analysis of the cranial NCC cluster reveals lineage/fate relationship between PM
and intermediate NCC subclusters 2' and 10'. Two intermediate NCC subclusters develop simultaneously and
independently from premigratory NCC. Apart from their cell cycle status, early migratory NCC formed from the
different intermediate subclusters are transcriptionally indistinguishable. Color coding of the cell population is
consistent with the re-clustering of the cranial NCC cluster into five smaller subclusters at a resolution of 0.26 as
shown previously in *Figure 2A*.

express S phase cell cycle genes such as *Pcna*, *Pol1* subunits, *Plk1*, *Ccnd,* and *Mcm* family members. In
contrast, NCC in subcluster 10' predominantly express G2/M phase cell cycle genes including *Mik67*,
*Aurka/b*, *Cenp,* and *Kif* family members (*Figure 3A and B*).

To understand the potential for any temporal or lineage relationships between the two interme-
diate stage NCC populations based on their different cell cycle states, we performed pseudotime
trajectory analysis on the entire NCC population using Monocle 3 (*Figure 3C and D*). Premigratory
and EMT intermediate NCC were identified as the earliest discrete populations to form among the
entire cranial NCC population (*Figure 3C*). In contrast, late migratory NCC represent a more mature
stage of NCC development (*Figure 3C*). Results from the pseudotime analysis indirectly support the
identities previously assigned to the cranial NCC subclusters under both resolutions. The trajectory
analysis also demonstrates that NCC can arise through two independent paths and initially become
two distinct intermediate populations (subclusters 2' and 10') during EMT (*Figure 3D*). Later the

trajectories or lineages of the intermediate NCC subclusters then merge back together into a single or common population of early migratory NCC, before ultimately maturing into still as yet undifferentiated late migratory NCC as they colonize the frontonasal or pharyngeal arch mesenchyme (*Figure 3D*). These results imply that NCC representing two distinct intermediate stages form simultaneously and independently during development. Moreover, the different cell cycle status of subclusters 2' and 10' suggest that premigratory NCC can undergo EMT and delamination in either S phase or G2/M phase of the cell cycle. The trajectory analysis further depicts that EMT intermediate NCC and their immediate lineages are not fate restricted to any specific cranial NCC derivative at this timepoint.

## Cell cycle regulation is critical for mouse cranial NCC EMT and delamination

Since EMT intermediate NCC are either in S phase or G2/M phase of the cell cycle, we then investigated whether cell cycle regulation plays a significant role in driving mouse cranial NCC delamination. We dissected E8.5 Wnt1-Cre;Rosa$^{LSL-eYFP}$ mouse embryos and examined the expression of cell cycle markers to compare the cell cycle status of delaminating NCC at the neural plate border in cranial tissues versus premigratory non-delaminating NCC in the neural plate border of the trunk. EdU and phospho-histone H3 (pHH3) were used to label S phase and G2/M phase of the cell cycle respectively (*Figure 4A*). A majority of delaminating cranial NCC express either EdU or pHH3 or both. Only a very small percentage of cells do not express either of these cell cycle markers (*Figure 4C*). In contrast, almost 50% of premigratory, non-delaminating trunk NCC in the dorsolateral neural tube do not express either cell cycle marker (*Figure 4D*). These observations imply that specific cell cycle phases are intimately connected to mouse cranial NCC EMT and delamination.

To further validate the association between cell cycle status and cranial NCC EMT and delamination, we inhibited S phase during early NCC development by incubating E8.0 CD1 mouse embryos in whole embryo roller culture with Aphidicolin. After 12 hr of treatment, we quantified the number of migratory NCC via Sox10 immunostaining to determine the number of premigratory NCC that delaminated. Cell cycle status was also evaluated in DMSO- (control) and Aphidicolin-treated samples via EdU and pHH3 staining. Based on our trajectory analyses, we hypothesized that inhibiting S phase progression would block S phase delamination but not G2/M phase delamination. As we expected, the EdU signal was completely absent in Aphidicolin-treated embryos demonstrating that cells cannot enter S phase post treatment (*Figure 4—figure supplement 1*). The Aphidicolin treatment did not induce cell death as the level of TUNEL staining in treated embryos was similar to that observed in controls (*Figure 4—figure supplement 1*). Quantification of Sox10 positive cells revealed significantly fewer migratory NCC in the craniofacial region of Aphidicolin-treated embryos compared to DMSO-treated control embryos (*Figure 4E*). Similarly, we also treated E8.0 Mef2c-F10N-LacZ embryos with Aphidicolin for 12 hr in roller culture and observed fewer migratory NCC compared to control embryos as evidenced by LacZ staining (*Figure 4—figure supplement 1*). Interestingly, pHH3 is expressed by a major proportion of the remaining migratory NCC after Aphidicolin treatment (*Figure 4B*). This data is consistent with the pseudotime trajectory analysis that cranial NCC delamination in G2/M phase of the cell cycle (subcluster 10' NCC) is independent of delamination in S phase of the cell cycle (subcluster 2' NCC). Disrupting S phase of the cell cycle didn't prohibit EMT intermediate NCC in G2/M phase of the cell cycle from delaminating and forming migratory NCC that express G2/M phase cell cycle markers. Thus, our data shows that cranial NCC delamination is disrupted upon S phase cell cycle inhibition, which supports the hypothesis that cell cycle regulation is critical for cranial NCC delamination in mouse embryos.

## Spatiotemporal localization of intermediate stage NCC in vivo

To identify and define EMT intermediate stage NCC in vivo during mouse cranial NCC development, we used the scRNA-seq data to extract genes that were differentially expressed (threshold based on average logFC ≥0.25) in the intermediate NCC populations (*Figure 5—figure supplement 1*). Among the potential marker genes, we then selected *Dlc1*, *Sp5*, and *Pak3* based on their relatively high expression levels and specificity in cranial NCC, and more importantly, intermediate NCC populations (*Figure 5A*; *Figure 5—figure supplement 1*). During cranial NCC development, *Sp5* and *Pak3* are expressed at high levels in both premigratory and intermediate stage NCC. In contrast, *Dlc1* is highly expressed in intermediate and migratory NCC (*Figure 5A*). To further distinguish between

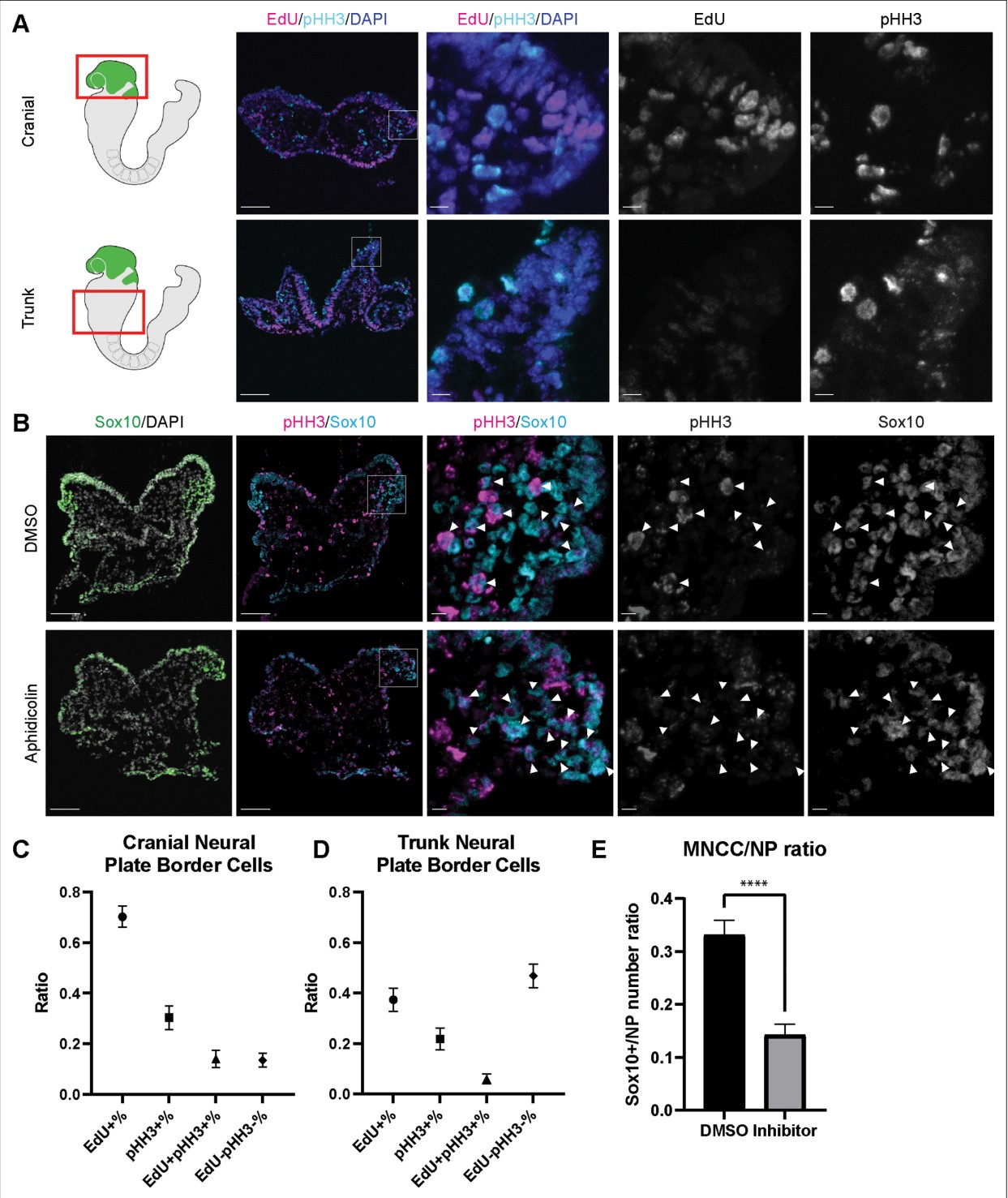

**Figure 4.** Cell cycle regulation plays an important role in mouse cranial neural crest cell (NCC) delamination and epithelial to mesenchymal transition (EMT). (**A**) Cell cycle marker staining of early E8.5 mouse embryonic cranial and trunk tissues reveals differences in cell cycle status between cranial delaminating premigratory NCC and trunk neural plate border cells. EdU (magenta) and phospho-histone H3 (pHH3) (cyan) staining were performed on 10 µm transverse sections of early E8.5 (5–7 somites) Wnt1-Cre;Rosa$^{LSL-eYFP}$ mouse embryo cranial and trunk tissues. (**B**) E8.0 CD1 mouse embryos treated with Aphidicolin exhibit reduced migratory NCC that primarily express pHH3. Cranial sections of treated embryos were stained with Sox10, EdU, and pHH3 (magenta) and arrowheads indicate migratory NCC expressing pHH3. Most remaining migratory NCC in Aphidicolin-treated samples express pHH3. In contrast, a small proportion of migratory NCC in control DMSO-treated samples express pHH3. (**C**) Cell cycle staining quantification of delaminating premigratory NCC in the cranial neural plate border shows that most cells express cell cycle markers. Staining and quantification

*Figure 4 continued on next page*

*Figure 4 continued*

were performed on delaminating premigratory NCC in the cranial neural plate border of 5–7 somite Wnt1-Cre;Rosa^LSL-eYFP mouse embryos (n=3). The neural plate border region was manually selected in the most dorsolateral domain of the neural plate. EdU+%=the percentage of EdU positive cells within eYFP positive delaminating premigratory NCC in the selected neural plate border domain. pHH3+%=the percentage of pHH3 positive cells within eYFP positive delaminating premigratory NCC. EdU +pHH3+%=the percentage of EdU and pHH3 double positive cells within eYFP positive delaminating premigratory NCC. EdU-pHH3-%=the percentage of EdU and pHH3 double negative cells within eYFP positive delaminating premigratory NCC. (**D**) Cell cycle staining quantification of trunk neural plate border cells shows that a significant proportion of cells do not express any cell cycle markers. Staining and quantification were performed on trunk neural plate border cells of 5–7 somite Wnt1-Cre;Rosa^LSL-eYFP mouse embryos (n=3). The neural plate border region was manually selected in the most dorsolateral domain of the neural plate. EdU+%=the percentage of EdU positive cells within DAPI positive neural plate border cells at the trunk axial level. pHH3+%=the percentage of pHH3 positive cells within DAPI positive trunk neural plate border cells. EdU+pHH3+%=the percentage of EdU and pHH3 double positive cells within DAPI positive trunk neural plate border cells. EdU-pHH3-%=the percentage of EdU and pHH3 double negative cells within DAPI positive trunk neural plate border cells. (**E**) Quantification of Sox10 expressing migratory NCC upon Aphidicolin and control treatment reveals fewer cranial migratory NCC in Aphidicolin-treated embryos. Sox10 staining and quantification were performed on cranial sections of 4–6 somite CD1 mouse embryos post treatment (n=3 per treatment; ****p<0.0001). For quantification, we calculated the ratio of Sox10 positive migratory NCC over DAPI positive neural plate/neuroepithelial cells. Scale bars: embryo sections 100 µm; section insets 10 µm.

The online version of this article includes the following figure supplement(s) for figure 4:

**Figure supplement 1.** Aphidicolin treatment on Mef2c-F10N-LacZ embryos shows consistent results as CD1 embryos.

premigratory, intermediate, and migratory NCC, *Wnt1* (a premigratory NCC marker) and *Sox10* (a migratory NCC marker) were also included in the in situ identification analyses alongside *Dlc1*, *Sp5*, and *Pak3* (**Figure 5A**).

To confirm that the intermediate stage markers *Dlc1*, *Sp5*, and *Pak3* are expressed during NCC delamination, and determine the spatial location of the intermediate stage NCC, we performed SABER-FISH. SABER-FISH was chosen for our multiplexed analyses of gene expression because SABER-FISH probes lack secondary structure which facilitates increased sensitivity and depth of tissue penetration. SABER-FISH oligo pools were designed for *Wnt1*, *Sox10*, *Dlc1*, *Sp5*, and *Pak3*, using stringent parameters (**Kishi et al., 2019**). Intermediate stage marker probes were validated by comparing the expression patterns of the SABER-FISH staining for *Dlc1*, *Sp5*, and *Pak3* with traditional in situ hybridization staining in transverse histological sections (**Figure 5—figure supplement 2**). The expression of *Dlc1*, *Sp5*, and *Pak3* matched between SABER-FISH and traditional in situ hybridization methods, validating our probe design (**Figure 5—figure supplement 2**).

Following individual validation, we then performed combined staining to visualize the spatial distribution of all the genes in the same tissue section (**Figure 5B**). The intermediate stage markers *Dlc1*, *Sp5,* and *Pak3* appeared to overlap in expression in the dorsal most region of the neural fold, where EMT takes place as evidenced by the presence of *Sox10*-labeled migratory NCC adjacent to the neuroepithelium (**Figure 5B**). Spatial expression of these markers changes over developmental time, thus overlap of expression in the dorsal most region of the neural fold was not observed in older embryos in regions where EMT had concluded. To better visualize and confirm co-localized expression of these genes in the same dorsolateral region of the neural plate border, we generated polyline kymographs depicting the average intensity of each gene's fluorescent signal along the dorsal most region of the neural fold and into the migratory NCC population (**Figure 5C**). We observed a consistent pattern of activity in which *Wnt1* is highly expressed in the dorsal neuroepithelium (**Figure 5C**). However, in the most dorsolateral domain, where *Wnt1* expression is slightly diminished, the intermediate stage NCC markers *Dlc1*, *Sp5*, and *Pak3* are highly expressed (**Figure 5C**). In contrast, minimal *Sox10* expression is detected in this transition region at the edge of the neuroepithelium, but high levels of *Sox10* in the clear absence of *Wnt1*, *Dlc1*, *Sp5*, and *Pak3*, is observed in migratory NCC located more ventrally (**Figure 5C**). Therefore, our data indicates that EMT intermediate stages can not only be transcriptionally defined, but also spatially resolved to the dorsolateral most region of the neuroepithelium.

## EMT intermediate stage marker gene Dlc1 regulates NCC delamination

Having transcriptionally defined intermediate stage NCC and determined their spatial location during delamination, it was important to test whether any of the intermediate stage signature genes, *Dlc1*, *Pak3*, or *Sp5*, play functional or essential roles in NCC development. We prioritized *Dlc1* over *Pak3* and *Sp5* because *Dlc1* is not expressed by premigratory NCC but is expressed at high levels in all EMT

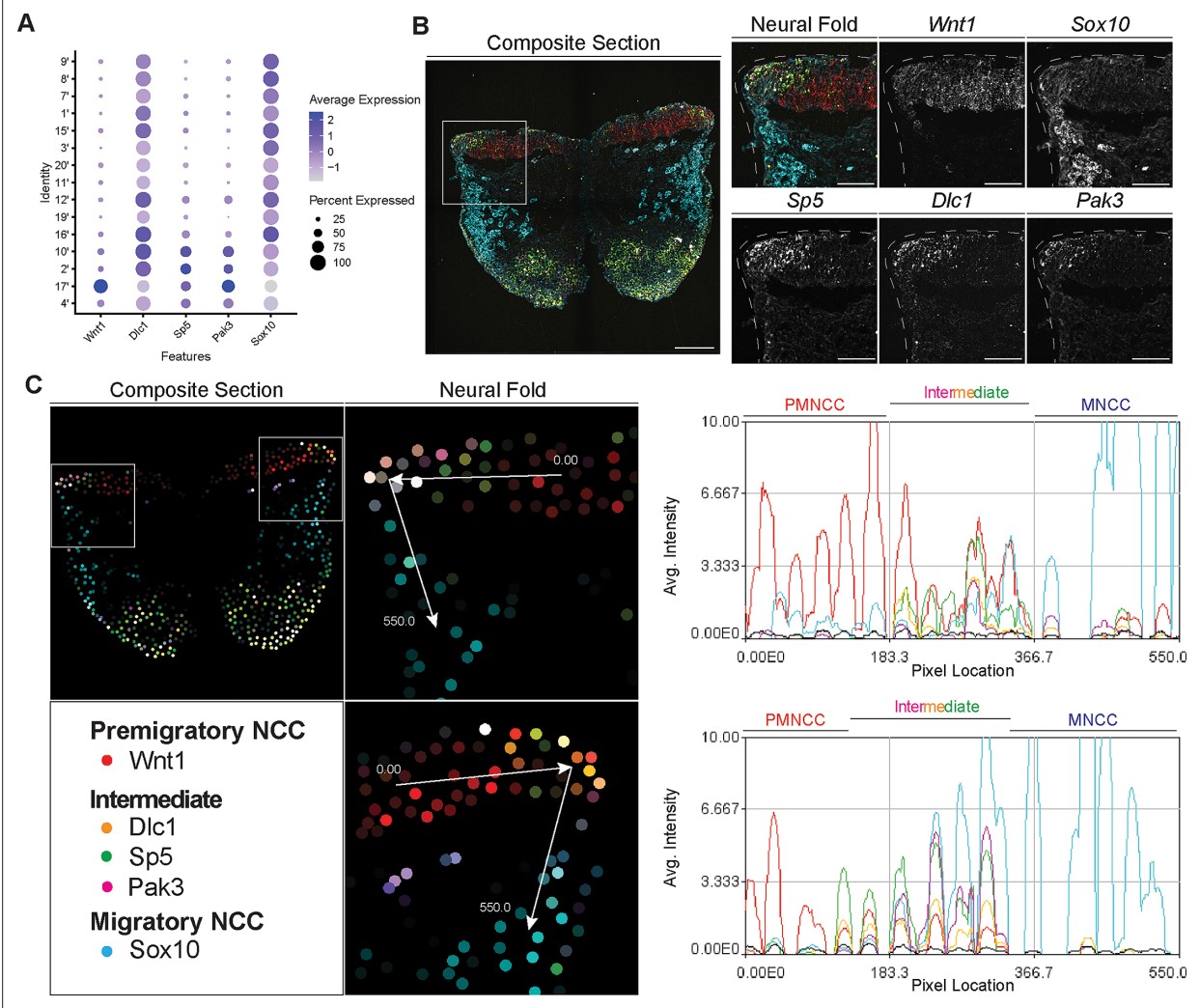

**Figure 5.** Signal amplification by exchange reaction for multiplexed fluorescent in situ hybridization (SABER-FISH) of epithelial to mesenchymal transition (EMT) intermediate stage markers pinpoints the location of EMT intermediate neural crest cell (NCC) within the dorsal most region of the neural fold. (**A**) Dotplot showing the expression of selected EMT intermediate NCC markers in early migratory NCC subclusters (resolution 2.0). (**B**) SABER-FISH staining of premigratory, EMT intermediate stage, and migratory NCC marker genes on the same section. Higher magnification insets of the left side neural fold (box) showing that *Wnt1* is expressed in the neuroepithelium and *Sox10* is expressed in migratory NCC populating the underlying mesenchyme. *Dlc1*, *Sp5*, and *Pak3* are expressed in the dorsolateral most region of the neuroepithelium. (**C**) 2D map showing the number of transcripts per cell, calculated from the SABER-FISH staining. To evaluate the expression of each gene within and across tissues, a polyline kymograph was generated along the track indicated by the arrows at a width of 100 pixels. The polyline kymograph can be seen to the right of each neural fold map it depicts. At the beginning of the track, *Wnt1* expression is highest, demarcating the dorsal lateral domain of the neuroepithelium. Toward the middle of the track, at the location of the most dorsolateral region of the neuroepithelium, *Wnt1* is expressed along with the intermediate stage markers *Dlc1*, *Sp5*, and *Pak3*. As the track progresses to just outside of the neuroepithelium, *Sox10* expression appears and increases as the track continues through the migratory NCC population. Scale bars: embryo sections 100 µm; section insets 50 µm.

The online version of this article includes the following figure supplement(s) for figure 5:

**Figure supplement 1.** Expression of epithelial to mesenchymal transition (EMT) intermediate neural crest cell (NCC) markers in single-cell RNA sequencing (scRNA-seq) data.

**Figure supplement 2.** Expression of intermediate neural crest cell (NCC) markers *Dlc1*, *Sp5,* and *Pak3* by signal amplification by exchange reaction for multiplexed fluorescent in situ hybridization (SABER-FISH) and traditional in situ hybridization in E8.5 mouse embryos and cranial sections.

intermediate stage NCC. Furthermore, *Dlc1* null mutant mice are embryonically lethal and exhibit craniofacial malformation phenotypes (*Sabbir et al., 2010*). Specifically, gross morphological anomalies of craniofacial tissues such as frontonasal prominence and pharyngeal arch hypoplasia are apparent in E10.5 *Dlc1* null mouse embryos, which is suggestive of a perturbation of NCC development (*Sabbir*

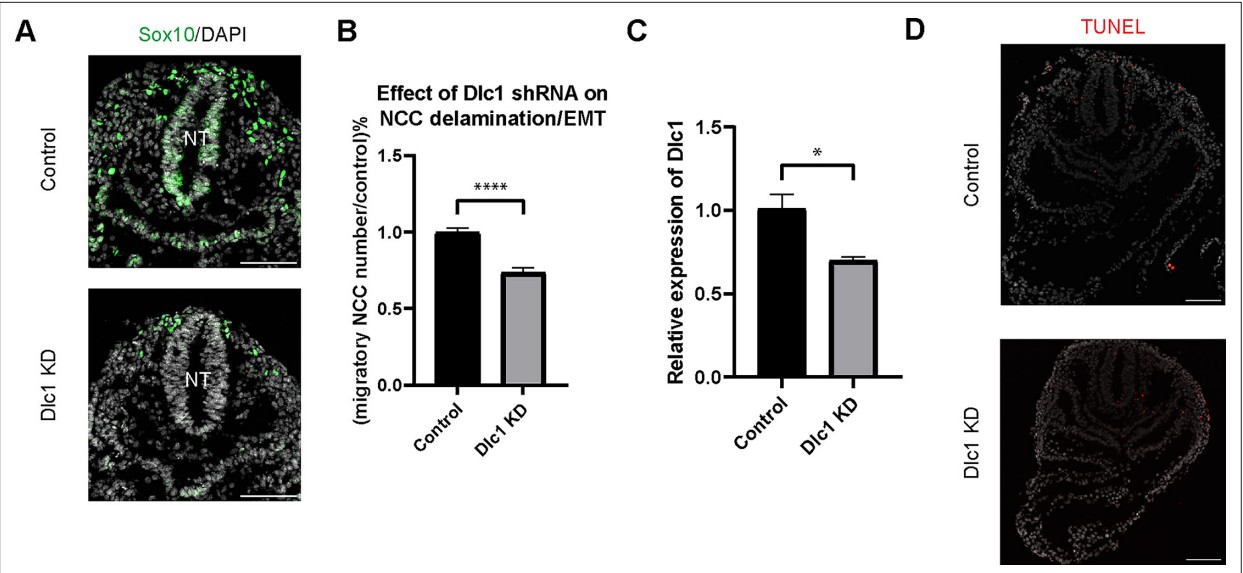

**Figure 6.** Dlc1 plays a regulatory role in mouse cranial neural crest cell (NCC) epithelial to mesenchymal transition (EMT) and delamination. (**A**) Sox10 immunostaining was performed on cranial sections of E8.5 control and *Dlc1* knockdown mouse embryos. (**B**) *Dlc1* knockdown significantly reduced the number of migratory NCC compared to the control. The number of Sox10+ migratory NCC was quantified in control (n=4) and all *Dlc1* knockdown (n=12) embryos. All datapoints in *Dlc1* knockdown samples were normalized to the control samples. ****p<0.0001. (**C**) *Dlc1* shRNA-based lentiviruses achieved an average of 30% reduction of *Dlc1* expression in all *Dlc1* knockdown embryos based on quantitative reverse transcription-PCR (qRT-PCR) analysis. *p<0.05. (**D**) TUNEL staining showed minimal cell death in *Dlc1* knockdown samples. Scale bars: embryo sections 100 μm.

The online version of this article includes the following figure supplement(s) for figure 6:

**Figure supplement 1.** *Dlc1* plays a regulatory role in mouse cranial neural crest cell (NCC) epithelial to mesenchymal transition (EMT) and delamination.

*et al., 2010*). In contrast, *Pak3* and *Sp5* are expressed in premigratory NCC, but in only 50–60% of intermediate stage NCC (*Figure 5A*). Moreover, *Pak3* and *Sp5* null mutant mice are healthy and fertile with no obvious abnormalities. Therefore, we hypothesized that alone, *Dlc1* loss-of-function would more likely perturb cranial NCC delamination.

To test this hypothesis, we knocked down *Dlc1* by injecting *Dlc1* shRNA-based lentiviruses and control scrambled shRNA lentiviruses into the amniotic cavity of E7.5 CD1 mouse embryos. Since the neural plate remains open at this developmental stage, all neuroepithelial cells are exposed or in contact with amniotic fluid containing virus. The embryos were then cultured for 24 hr after which we assessed the number of migratory NCC via *Sox10* staining, to determine how many premigratory NCC underwent EMT and delamination (*Figure 6A*). We subsequently observed that the number of migratory NCC was significantly reduced in all *Dlc1* knockdown embryos (*Figure 6B*). In fact, each of the *Dlc1* shRNA constructs led to a significant reduction in the number of migratory NCC compared to their respective control (*Figure 6—figure supplement 1*). Importantly, we did not observe any difference in cell death in *Dlc1* knockdown embryos comparable to controls (*Figure 6D*). On average, a 30% reduction of *Dlc1* expression was achieved by each *Dlc1* shRNA lentivirus knockdown (*Figure 6C*; *Figure 6—figure supplement 1*), and notably, the *Dlc1* isoforms targeted by the different *Dlc1* shRNA constructs used in this study suggest a correlation with *Dlc1* null mouse embryos and their phenotypes.

The Dlc1 splice-variant a, shRNA construct targets exon 1 specifically in *Dlc1* mRNA variant 2. Consequently, Dlc1 splice-variant a, shRNA is capable of exclusively eliminating the expression of *Dlc1* mRNA variant 2 (isoform 2) since the same exon region is not present in variant 1 or 3. Interestingly, one of the *Dlc1* null mouse models, *Dlc1*[gt/gt], carries a gene trap vector inserted into intron 1, which results in the reduction of the 6.1 kb transcript (Dlc1 isoform 2) alone (*Sabbir et al., 2010*). Therefore, it is possible that the craniofacial phenotypes observed in *Dlc1*[gt/gt] null mice are caused by abnormal cranial NCC EMT and delamination as shown by Dlc1 splice-variant a, knockdown in our data. In

contrast, Dlc1 splice-variant b, and Dlc1 splice-variant c, shRNA constructs both target exon 5 of *Dlc1* mRNA variants 2 and 3, which is the same as exon 9 of variant 1. Consequently, all three *Dlc1* variants should be diminished by Dlc1 splice-variant b, and Dlc1 splice-variant c, shRNAs. In another *Dlc1* null mouse model, exon 5 was deleted by replacing it with a neomycin resistance gene, which caused a reading frame shift and subsequently premature translation termination (*Durkin et al., 2005*). This led to the synthesis of truncated polypeptides containing only the first 77 amino acids, which encode the sterile alpha motif protein interaction domain and 23 novel residues. Since *Dlc1*$^{-/-}$ null, Dlc1 splice-variant b, and Dlc1 splice-variant c, shRNAs all cause disruptions in exon 5/9 of *Dlc1* transcripts, there is a strong correlation between the *Dlc1*$^{-/-}$ null craniofacial phenotypes and cranial NCC EMT defects observed in Dlc1 splice-variant b, and Dlc1 splice-variant c, knockdown mouse embryos. Collectively, these results therefore demonstrate an important functional role for Dlc1 in mammalian NCC EMT and delamination. Moreover, our results suggest that genes that are differentially expressed in intermediate NCC can play a regulatory role during EMT and delamination.

## Discussion

EMT is a cellular process that converts epithelial cells to mesenchymal cells. EMT is essential for normal development and is a key driver of disease pathogenesis, particularly cancer metastasis. Although classically considered to be a binary event, studies of EMT in cancer identified multiple intermediate states within the EMT spectrum, a phenomenon termed EMP. However, it remained to be determined whether developmental EMT is also a developmental EMP process. Our goal therefore was to determine whether intermediate stages of NCC development during EMT could be transcriptionally and spatially defined, and then test whether transitional stage-associated genes are functionally required for NCC EMT and delamination.

Through scRNA-seq analysis of mouse cranial tissues at E8.5, which coincided with the onset of NCC EMT and delamination, we identified two populations of NCC, whose gene expression profiles or signatures were representatives of intermediate stages between premigratory and migratory NCC. Furthermore, we determined that the two intermediate populations could be defined by their distinct transcriptional states which were consistent with being in either S phase or G2/M phase of the cell cycle. Although it is possible that the two intermediate populations represent a single population that is temporarily bifurcated due to cell cycle asynchrony, the pseudotime trajectory analysis suggests that these intermediate stage cranial NCC populations can undergo EMT and delaminate in either S phase or G2/M phase, simultaneously, and independently of each other. This is further supported by the observation that S phase cell cycle inhibition failed to completely eliminate NCC delamination since NCC delaminating in G2/M phase were observed. The two intermediate populations later merge into a single or common pool of early migratory NCC suggesting they do not have distinct fates following migration that can be tied to their cell cycle status as the time of delamination. This is not indicative of a common progenitor of both ectomesenchyme and neuro/glial/pigment derivatives, but rather, is consistent with the known plasticity and potency of migrating NCC (*Sandell and Trainor, 2006*; *Trainor and Krumlauf, 2000a*; *Trainor and Krumlauf, 2000b*; *Trainor and Krumlauf, 2001*).

These results correlate with observations that cell cycle status is also a critical factor regulating NCC delamination in avian and zebrafish embryos. For example, BrdU incorporation, which demarcates proliferating cells in S phase of the cell cycle, was previously used to evaluate the cell cycle status of emigrating trunk NCC, dorsal midline neuroepithelial cells, and surrounding cells at the segmental plate, epithelial somite and dissociating somite axial levels in chicken embryos (*Burstyn-Cohen and Kalcheim, 2002*). Most emigrating trunk NCC (about 80%) at the epithelial somite and dissociating somite axial levels were in S phase of the cell cycle, while less than 50% of dorsal neuroepithelial cells were BrdU+. Similarly, slice culture of the trunk of chicken embryos also revealed that most premigratory NCC contained basally positioned nuclei indicative of S phase. Interestingly, however, a small proportion of premigratory NCC presented as round mitotic cells, whose daughter cells later became migratory (*Ahlstrom and Erickson, 2009*). Furthermore, in vivo time-lapse imaging of chicken embryos revealed that half of the delaminating trunk NCC that were tracked displayed signs of cell division (*McKinney et al., 2013*). However, in most cases, only one progeny of a mitotic

premigratory NCC was observed to exit the neural tube and become a migratory NCC. Interestingly, time-lapse imaging of zebrafish embryos also revealed active cell division in dorsal neuroepithelial cells prior to NCC delamination and EMT. The daughter cells of those divisions then translocate into the basal side of the neuroepithelium, where EMT subsequently occurs (*Berndt et al., 2008*). Whether delaminating NCC undergo proliferation and cell division in zebrafish embryos remains to be investigated, however, active cell division in the dorsal neuroepithelium is a shared feature of NCC delamination in avian and aquatic species. Even though our findings primarily illustrate the association of distinct cell cycle phases with intermediate stage NCC during EMT and delamination, this work has emphasized the importance of further examining the cell division and cell cycle activities of delaminating NCC in mouse embryos as critical contributors to normal development and the pathogenesis of neurocristopathies.

In contrast to delaminating trunk NCC, only around 30% of delaminating cranial NCC in chicken embryos were found to be in S phase (*Théveneau et al., 2007*) illustrating considerable differences between cranial and trunk NCC and in the correlation between cell cycle phase and delamination. Blocking G1/S transition in the trunk of chicken embryos via in ovo electroporation, or via small molecule inhibitors in explanted neural primordia, prevents the onset of NCC delamination (*Burstyn-Cohen and Kalcheim, 2002*). Furthermore, BMP and Wnt canonical signaling regulates the G1/S transition and promotes trunk NCC delamination (*Burstyn-Cohen et al., 2004*). Although it remains to be determined which signaling pathways regulate cell cycle phase progression in the intermediate or transitional populations of cranial NCC in mouse embryos, our results illustrate an evolutionarily conserved mechanistic role for cell cycle progression in NCC delamination in vertebrate embryos.

The intermediate populations of cranial NCC in mouse embryos exhibited transcriptional profiles that were characterized by the downregulation of tight junction and polarity genes. This is consistent with the breakdown of intercellular tight junctions and degradation of apicobasal polarity, which are hallmarks of EMT (*Zhao and Trainor, 2023*). Further interrogation of genes that were differentially expressed in the intermediate NCC populations revealed *Dlc1*, *Sp5,* and *Pak3* based on their relatively high expression levels as potentially specific markers, and regulators of intermediate NCC populations. Through SABER-FISH staining of *Dlc1*, *Sp5*, and *Pak3* in combination with *Wnt1* as a marker of premigratory NCC, and *Sox10* as a marker of migratory NCC, we spatially resolved the location of intermediate NCC to the most dorsolateral domain of the cranial neural plate in E8.5 mouse embryos. We then prioritized *Dlc1* for functional analyses because it is expressed at high levels in all EMT intermediate stage NCC, but not in premigratory NCC. Lentiviral shRNA knockdown of *Dlc1* in cultured mouse embryos resulted in a significant reduction in the number of migratory NCC, which may account for the craniofacial and cardiac malformation phenotypes observed in *Dlc1* null mutant mice (*Sabbir et al., 2010*). Gross morphological defects in craniofacial tissues are apparent in E10.5 *Dlc1* null mouse embryos. More specifically, *Dlc1* null embryos exhibit hyperplastic frontonasal prominences and pharyngeal arches, a neural tube closure defect and underdeveloped atrial and ventricular walls of the heart. Although phenotypes at earlier developmental stages and during NCC development were not examined in these mutants, *Dlc1* is primarily expressed in intermediate stage NCC, and plays an important role in NCC EMT, delamination, and the subsequent development of NCC-derived tissues. Dlc1 is a Rho GTPase-activating protein (GAP) that regulates the activity of Rho family GTPases Rho and Cdc42 (*Kim et al., 2008*). RhoGTPases are known to regulate cell morphology and motility through modulating the activity of the actin cytoskeleton. More specifically, Rho has been shown to facilitate the formation of stress fibers, while Cdc42 is involved in filopodium formation (*Kim et al., 2008*). RhoGTPases also regulate the organization of tight junctions, which breakdown during EMT (*Popoff and Geny, 2009*; *Terry et al., 2010*). Therefore, the effect of *Dlc1* knockdown on cranial NCC EMT and delamination may be mediated via disrupted RhoGTPase activity and subsequent downstream cellular changes.

A similar regulatory role for Dlc1 and other GAP family members has also been observed during NCC EMT and delamination in chicken and zebrafish embryos. In chicken embryos, *Dlc1* overexpression results in ectopic trunk NCC delamination, including apically into the neural tube lumen, due to a disruption in apical-basal polarity of dorsal neuroepithelial cells (*Liu et al., 2017*). Furthermore, NCC overexpressing *Dlc1* exhibit a loss of directionality during migration. Conversely, *Dlc1* inhibition and

depletion results in less NCC emigration and thus fewer migratory NCC. In addition, the downregulation of *Dlc1* in migrating NCC restricts their motility. Thus, *Dlc1* regulates trunk NCC delamination and migration in chicken embryos. Interestingly, in zebrafish embryos, another GAP family member Arhgap has been shown to modulate NCC EMT and delamination via the localization of Rho activation to designated subcellular compartments in concert with promoting localized actomyosin contraction to trigger directional cell motility (*Clay and Halloran, 2013*). More specifically, the knockdown of Arhgap in NCC results in Rho activation, which in turn inhibits NCC EMT and delamination. These data suggest that the role of Dlc1 during NCC delamination might be evolutionarily conserved in vertebrate embryos. However, whether Dlc1 regulates mouse cranial NCC delamination through localized activation of Rho remains to be determined. Nevertheless, consistent with similarities in EMT, delamination, and cell migration between NCC and cancer cells, Dlc1 may also play an important role in promoting cell migration during cancer progression. High levels of *DLC1* expression are detected in most melanoma tissues, and functional studies have revealed that DLC1 is both necessary and sufficient for melanoma growth and metastasis (*Yang et al., 2020b*).

Our study focused on the identification and characterization of NCC EMT intermediate states and essential regulatory factors during cranial NCC delamination and EMT. Nevertheless, whether trunk NCC delamination and EMT also involve intermediate stages and are regulated by similar genetic/cellular elements remains to be explored. Differences between cranial and trunk NCC delamination have been previously identified. For example, in mouse embryos, cranial NCC delaminate from the dorsolateral neuroepithelium or neural plate border prior to neural plate closure. Trunk NCC, on the other hand, delaminate from the dorsal neural tube after neural plate closure. In the cranial region, premigratory NCC delaminate in bulk as a collective stream, to give rise to the cranial ganglia and craniofacial bones and cartilages. In the trunk region, premigratory NCC delaminate progressively one after another in a chain migration fashion to form the sympathetic ganglia, the dorsal root ganglia, glial cells, and melanocytes. Additionally, key signaling pathways and regulatory networks governing cranial and trunk NCC in various species have also been shown to differ in these distinct axial populations of NCC (*Theveneau and Mayor, 2012*). Due to these differences between cranial and trunk NCC, the molecular and cellular mechanisms discovered during cranial NCC delamination and EMT do not necessarily represent the signaling scheme in trunk NCC. To enhance our knowledge of the gene regulatory network governing trunk NCC delamination/EMT and to decipher whether EMT intermediate states also exist in the trunk region of mouse embryos, scRNA-seq must be performed using trunk tissues during trunk NCC delamination. Since trunk NCC delamination/EMT occurs progressively from anterior to posterior as the embryos grow, it is very important to select correct axial domains of the trunk to match the appropriate or required developmental stage.

In conclusion, through scRNA-sequencing we have transcriptionally identified two distinct intermediate stages of NCC during EMT and delamination based primarily on cell cycle status. Delamination in S phase or G2/M phase seems to occur simultaneously but also independently, resulting in a single or common pool of early migratory NCC. Further interrogation of our transcriptomic dataset revealed *Dlc1* to be a key molecular marker of intermediate stage NCC, and their location in situ in the dorsolateral neural plate, which we spatially resolved in E8.5 mouse embryos. Lastly, we tested and functionally validated that *Dlc1* plays an important role in NCC delamination in mouse embryos. Taken together, our identification and characterization of intermediate stage cranial NCC during their delamination are consistent with NCC EMT being a developmental EMP event. Similar to intermediate EMT states in cancer metastasis, NCC downregulate certain epithelial cell features but maintain co-expression of epithelial cell markers and mesenchymal cell markers during EMT and delamination. Additionally, intermediate stage NCC are localized at the dorsolateral edge of the neural plate border, which is reminiscent of the localization of intermediate EMT states at the leading edge of invasion in several types of primary tumors. However, unlike certain intermediate EMT cells present in the lymph nodes or circulating tumor cells, mouse EMT intermediate NCC represent a transient state and eventually form migratory NCC with mesenchymal character. This suggests that EMP may be a more common developmental phenomenon. Our transcriptional data and signatures of intermediate stage NCC during EMT and delamination can serve as a useful resource for the community. This also now sets the stage for uncovering the gene regulatory networks that govern intermediate stage NCC development and function, and for exploring whether EMP is a feature of other developmental and pathological EMT events such as in gastrulation, wound healing, and fibrosis.

# Materials and methods

**Key resources table**

| Reagent type (species) or resource | Designation | Source or reference | Identifiers | Additional information |
|---|---|---|---|---|
| Gene (*Mus. musculus*) | Dlc1 | GenBank | MGI:MGI:1354949 | |
| Strain, strain background (*M. musculus*) | Wnt1-Cre | The Jackson Laboratory | RRID:IMSR_JAX:003829 | |
| Strain, strain background (*M. musculus*) | RosaeYFP | The Jackson Laboratory | RRID:IMSR_JAX:006148 | |
| Strain, strain background (*M. musculus*) | Mef2c-F10N-LacZ | doi: https://doi.org/10.1016/j.ydbio.2015.02.022; **Aoto et al., 2015** | | |
| Cell line (*Homo sapiens*) | 293T | ATCC | CRL-3216 | |
| Biological sample (*M. musculus*) | Mouse embryonic cranial tissues | This paper | | Freshly isolated from E8.5 mouse embryos |
| Antibody | Anti-GFP (Rabbit polyclonal) | Invitrogen | Cat#: A-6455 | IF(1:500) |
| Antibody | Anti-GFP (Rabbit monoclonal) | Invitrogen | Cat#: G10362 | IF(1:500) |
| Antibody | Anti-rabbit IgG (H+L) Cross-Adsorbed Secondary Antibody, Alexa Fluor 488 (Goat polyclonal) | Invitrogen | Cat#: A-11008 | IF(1:500) |
| Antibody | DAPI | Sigma-Aldrich | Cat#: D9564 | IF(1:1200) SABER-FISH(1:1000) |
| Antibody | Anti-mouse SOX10 (Rabbit monoclonal) | Abcam | Cat#: ab155279 | IF(1:500) |
| Antibody | Anti-human phospho-Histone H3 (Mouse monoclonal) | Millipore | Cat#: 05–806 | IF(1:500) |
| Antibody | Anti-mouse secondary antibody Alexa Fluor 546 (Goat) | Invitrogen | Cat#: A21045 | IF(1:500) |
| Antibody | Anti-rat secondary antibody Alexa Fluor 647 (Goat) | Invitrogen | Cat#: A21267 | IF(1:500) |
| Antibody | Anti-Mouse IgG (H+L) Highly Cross-Adsorbed Secondary Antibody, Alexa Fluor Plus 647 (Donkey polyclonal) | Invitrogen | Cat#: #A32787 | IF(1:500) |
| Recombinant DNA reagent | Dlc1 (plasmid) | doi: https://doi.org/10.1016/j.febslet.2004.12.090 | | ISH; Dr. Marian Durkin |
| Recombinant DNA reagent | Pak3 (plasmid) | doi: https://doi.org/10.2337/db13-0384 | | ISH; Dr. Gerard Gradwohl |
| Recombinant DNA reagent | Sp5 (Plasmid) | doi: https://doi.org/10.1371/journal.pone.0087018 | | ISH; Dr. Terry Yamaguchi |

*Continued on next page*

*Continued*

| Reagent type (species) or resource | Designation | Source or reference | Identifiers | Additional information |
|---|---|---|---|---|
| Recombinant DNA reagent | shRNA to Dlc1 | GeneCopoeia | Cat#: MSH100727-LVRU6MP | |
| Sequence-based reagent | Dlc1_F | This paper | qRT-PCR primer | AGCGGCTGTGAAAGAAA |
| Sequence-based reagent | Dlc1_R | This paper | qRT-PCR primer | GCATTACCCTTGGAGAAGA |
| Sequence-based reagent | B2M_F | This paper | qRT-PCR primer | CACTGACCGGCCTGTATGC |
| Sequence-based reagent | B2M_R | This paper | qRT-PCR primer | GGTGGCGTGAGTATACTTGAATTTG |
| Sequence-based reagent | CANX_F | This paper | qRT-PCR primer | CCAGACCCTGATGCAGAGAAG |
| Sequence-based reagent | CANX_R | This paper | qRT-PCR primer | CCTCCCATTCTCCGTCCATA |
| Sequence-based reagent | Sox10 | other; *Kishi et al., 2019* | | Oligo pools for SABER-FISH designed and ordered through IDT using stringent settings as previously described |
| Sequence-based reagent | Wnt1 | other; *Kishi et al., 2019* | | Oligo pools for SABER-FISH designed and ordered through IDT using stringent settings as previously described |
| Sequence-based reagent | Dlc1 | other; *Kishi et al., 2019* | | Oligo pools for SABER-FISH designed and ordered through IDT using stringent settings as previously described |
| Sequence-based reagent | Sp5 | other; *Kishi et al., 2019* | | Oligo pools for SABER-FISH designed and ordered through IDT using stringent settings as previously described |
| Sequence-based reagent | Pak3 | other; *Kishi et al., 2019* | | Oligo pools for SABER-FISH designed and ordered through IDT using stringent settings as previously described |
| Commercial assay or kit | In Situ Cell Death Detection Kit, TMR red | Roche | #12156792910 | |
| Commercial assay or kit | Click-iT Plus EdU Cell Proliferation Kit for Imaging, Alexa Fluor 555 dye | Invitrogen | Cat#: C10638 | |
| Commercial assay or kit | Qiagen miRNeasy Micro Kit | Qiagen | Cat#: 217084 | |
| Commercial assay or kit | SuperScript III First-Strand Synthesis System | Invitrogen | Cat#: 18080051 | |
| Chemical compound, drug | Aphidicolin | Sigma-Aldrich | Cat#: A0781 | (0.5 ug/ml) |
| Chemical compound, drug | Ampicillin | Amresco | Cat#: 0339 | (100 ug/ml) |
| Chemical compound, drug | PerfeCTa SYBR Green FastMix | Quantbio | Cat#: 95072–250 | |
| Chemical compound, drug | Tissue Rinse Solution A | Millipore | Cat#: BG-8-A | |
| Chemical compound, drug | Tissue Rinse Solution B | Millipore | Cat#: BG-8-B | |

*Continued on next page*

*Continued*

| Reagent type (species) or resource | Designation | Source or reference | Identifiers | Additional information |
|---|---|---|---|---|
| Chemical compound, drug | Tissue Stain Base Solution | Millipore | Cat#: BG-8-C | |
| Software, algorithm | FIJI/ImageJ | Fiji | | https://imagej.net/software/fiji/ |
| Software, algorithm | RStudio | Posit | | https://posit.co/download/rstudio-desktop/ |
| Other | VECTASHIELD Antifade Mounting Medium with DAPI | Vector Laboratories | Cat#: H-1200 | |
| Other | Tissue-Tek O.C.T. Compound | VWR | Cat#: 25608–930 | |
| Other | Anti-Digoxigenin-AP, Fab fragments | Roche | 11093274910 | ISH(1:2000) |

## Mice and animal husbandry

All mice were kept in a 14 hr-light and 10 hr-dark light cycle in the Laboratory Animal Services Facility at the Stowers Institute for Medical Research. All animal experiments were conducted in accordance with Stowers Institute for Medical Research Institutional Animal Care and Use Committee (IACUC)-approved protocol (IACUC no. 2022-143). Wnt1-Cre mice (*H2afv*<sup>Tg(Wnt1-cre)11Rth</sup> Tg(Wnt1-GAL4)11Rth/J, JAX stock #003829) and Rosa<sup>LSL-eYFP</sup> mice (B6.129X1-Gt(ROSA)26Sortm1(EYFP)Cos/J, JAX stock # 006148) were obtained from the Jackson Laboratory and maintained and genotyped as previously described (*Chai et al., 2000*; *Jiang et al., 2000*). Wnt1-Cre was maintained as a heterozygous allele and crossed to homozygous Rosa<sup>LSL-eYFP</sup> to generate Wnt1-Cre;Rosa<sup>LSL-eYFP</sup>. Mef2c-F10N-LacZ mice, in which LacZ expression is regulated by an NCC-specific enhancer of *Mef2c*, were maintained and genotyped as previously described (*Aoto et al., 2015*).

## Immunohistochemistry

### Wholemount

Embryos were dissected and fixed in 4% PFA (in PBS) overnight. The next day the embryos were rinsed three times for 5 min in PBS, dehydrated through an increasing methanol series from 25% methanol/PBS to 100% methanol and stored at –20°C until needed. Embryos were incubated in Dent's bleach (4:1:1 methanol:DMSO:hydrogen peroxide) for 2 hr in the dark at room temperature, rinsed in 100% methanol for 15 min, and rehydrated through a decreasing methanol series from 100% methanol to PBT (0.1% Triton in 1× PBS). Embryos were blocked in 2% goat serum + 2% bovine serum albumin for 2 hr at room temperature before primary antibody (GFP, 1:500, Invitrogen #A6455) was added and the embryos were incubated overnight at 4°C with rocking. Embryos were washed three times for 5 min in PBT and then six times for 1 hr at room temperature with rocking. Embryos were incubated in secondary antibody (Alexa Fluor 488 Goat anti-Rabbit, 1:500, Invitrogen #11008; DAPI, 1:1200, Sigma-Aldrich #D9564) at 4°C in the dark overnight with rocking, then washed three times for 5 min in PBT and six times for 1 hr at room temperature with rocking. Embryos were then mounted in VECTA-SHIELD Antifade Mounting Medium with DAPI (Vector Laboratories #H-1200) for imaging.

### Cryosection

Embryos were dissected and fixed in 4% PFA (in PBS) overnight. Embryos were then transferred to 1× PBS and rocked for 10 min at 4°C, before being incubated in 30% sucrose in 1× PBS, at 4°C for 30 min to 5 hr with rocking (or until embryos were equilibrated as indicated by their sinking to the bottom of the tube). Embryos were embedded in Tissue-Tek O.C.T. Compound (VWR #25608-930), sectioned transversely at 10 µm thickness on an NX70 Cryostar cryostat, permeabilized three times for 5 min in PBT, and blocked in 2% goat serum + 2% bovine serum albumin for 2 hr at room temperature. Sections were then incubated in primary antibody overnight at 4°C and the following primary antibodies were used: Sox10 (1:500, Abcam #ab155279), pHH3 (1:500, Millipore #05-806), GFP (1:500, Invitrogen #G10362). Sections were then washed three times for 5 min in PBT at room temperature,

before diluted secondary antibody and/or conjugated primary antibody solutions were added and the sections were incubated for 2 hr in the dark at room temperature. Sections were washed three times for 5 min in PBT before mounting in VECTASHIELD. Secondary antibodies included Alexa Fluor 488 Goat anti-Rabbit (1:500, Invitrogen, #11008), 546 Goat anti-Mouse (1:500, Invitrogen, #A21045), 647 Goat anti-Rat (1:500, Invitrogen, #A21267), 647 Donkey anti-Mouse (1:500, Invitrogen, #A32787), DAPI (1:1200, Sigma-Aldrich, #D9564).

## TUNEL staining

Cryosections were washed three times for 5 min in PBT and then permeabilized in 0.1% sodium citrate/PBT for 5 min. Cryosections were then washed three times for 5 min in PBT before being incubated with 1:10 TUNEL enzyme buffer (Roche #12156792910) at 37°C in the dark for 2 hr. Sections were washed three times for 5 min in PBS prior to counterstaining with DAPI and/or mounting in VECTASHIELD.

## EdU labeling

EdU staining was performed on cultured whole mouse embryos. Briefly, complete media containing 50% DMEM-F12-Glutamax, 50% rat serum, and 1× penicillin/streptomycin was prewarmed at 37°C in roller culture bottles in a 5% $CO_2$, 5% $O_2$, and 90% $N_2$ atmosphere (*Muñoz and Trainor, 2019*; *Sakai and Trainor, 2014*). After E8.5 CD1 embryos were dissected in Tyrode's buffer with an intact yolk sac, and equilibrated in culture media for 30 min, EdU (Invitrogen #C10638) was added to the media according to the manufacturer's protocol at a working concentration of 500 µM. DMSO was added to control embryos. Embryos were then incubated for 15 min, after which they were briefly rinsed in warm equilibrated culture media followed by Tyrode's buffer before fixing in 4% PFA (in PBS) overnight at 4°C. Fixed embryos were embedded and sectioned at 10 µm, and then stained following the manufacturer's instructions, with fluorescent immunostaining performed as needed after EdU staining.

## Single-cell RNA sequencing

Tissue collection, cell processing and sequencing, and data processing were performed as previously described (*Falcon et al., 2022*). Briefly, six *Mef2c-F10N-LacZ* (*Aoto et al., 2015*) and six *Wnt1-Cre;RosaeYFP* (*Chai et al., 2000*) E8.5 mouse embryo littermates were collected. It's important to note we included at least two eYFP negative controls together with the *Wnt1-Cre;RosaeYFP* embryos, and negative controls of *Mef2c-F10N-LacZ*. Briefly, after isolating E8.5 embryos, we then dissected the head from those embryos, and performed scRNA-seq on dissociated cranial tissues. Cranial tissues were then dissociated into single cells, and 12,000–15,000 cells per sample were loaded on a Chromium Single Cell Controller (10x Genomics). Libraries were prepared using the Chromium Next GEM Single Cell 3' Library & Gel Bead Kit v3.1 (10x Genomics), quality control checked, and then pooled at equal molar concentrations and sequenced on an Illumina NovaSeq 6000 S1 flow cell. Raw sequencing data was processed using Cell Ranger (v3.0.0, 10x Genomics) and after mitochondria and other feature thresholding, the final dataset used for analysis consisted of 21,190 cells (12,498 cells for *Wnt1-Cre;RosaeYFP* and 8692 for *Mef2c-F10N-LacZ*) and 29,041 genes, and is available at the Gene Expression Omnibus (accession no. GSE168351). R (v3.6.1) was used for downstream analysis. The Seurat package (v3.1.5.9003) (*Stuart et al., 2019*) was used to normalize data via the SCTransform method (*Hafemeister and Satija, 2019*). For clustering, 3000 highly variable genes were selected, and the first 46 principal components based on those genes were used to identify 6 initial clusters at a resolution of 0.05 using the shared nearest neighbor method. Cranial NCC were identified as one of the six initial clusters based on the expression of tissue-specific marker genes. The cranial NCC cluster was then subdivided at resolution = 0.26 into five subclusters, three of which were characterized as the early migratory NCC based on NCC development gene expression. Early migratory NCC were further subdivided into 15 subclusters at resolution = 2.0 to explore the presence of EMT intermediate NCC.

## Pseudotime trajectory analysis

Trajectory analysis was performed using Monocle 3 (0.2.2) with closed loop set to true and use partition set to false on cranial NCC, which were subset from the single-cell dataset after doublet identification with DoubletFinder (2.0.2). The resulting pseudotime estimates were added back to the Seurat

metadata table for visualization and the trajectory was plotted on the original clusters identified in the Seurat analysis.

## SABER-FISH staining and imaging

All oligo pools, concatemer hairpins, and fluorophore probes were designed and ordered through IDT using stringent settings as previously described (*Kishi et al., 2019*). Only fluorophores 488, 594, and 647 were used to prevent overlap in spectral emission and channel bleed-through. SABER-FISH probes for each gene were made following the PER concatamerization protocol. The probes were allowed to elongate for 2 hr at 37°C before heat inactivation of the polymerase. A sample of the probes was run on a gel to confirm elongation length and ensure that no secondary products had formed in the process. Probes were then cleaned using the QIAGEN PCR Purification kit and the final concentration of the generated probes were measured by Nanodrop.

CD1 embryos were collected at the 5-6ss and fixed overnight in 4% PFA (in PBS) at 4°C. Embryos were then washed five times in PBS with DEPC-PBTW (1× DEPC-PBS + 0.1% Tween-20) and transferred into 30% sucrose PBTW and rocked overnight at 4°C. Next, embryos were embedded in OCT, frozen and cryosectioned at 20 µm thickness. Sections were placed on Histogrip-treated slides and warmed to encourage adherence, washed three times for 5 min in PBTW, incubated in hybridization wash buffer and allowed to equilibrate to 37°C (the lowest melting temperature limit out of our probe set). Sections were then incubated with pre-equilibrated probes overnight (roughly 16 hr) at 37°C. The next day the sections were washed with hybridization wash buffer, 2× SSCT (2× SSC + 0.1% Tween-20) and PBTW and then the first set of fluorophores were applied to the sections (set 1: Sox10, Wnt1, and Sp5; set 2: Pak3 and Dlc1) and left to hybridize at 37°C for 30 min. Following fluorophore probe hybridization, sections were washed in PBTW with 1:1000 DAPI for 10 min followed by PBTW two times for 5 min. The slides were then mounted in VECTASHIELD containing DAPI with a 1.5 glass coverslip.

Sections were imaged with a Nikon CSU-W1 inverted spinning disk equipped with an sCMOS camera. Each laser was set to 400 µs acquisition speed and a z-stack of 32 slices (1 µm per slice) was acquired. Following imaging of the first set of probes, the coverglass was gently removed from each slide. The first set of fluorophore probes were removed by washing three times for 5 min in PBTW washes, followed by three washes with displacement buffer and another three washes in PBTW. Slides were remounted with VECTASHIELD, covered with a coverslip and imaged to ensure all previous fluorescent probes had been removed. After imaging, the coverslip was removed again, and the sections were rinsed in PBTW three times for 5 min. The second fluorophore probes were then applied to the sections and allowed to hybridize for 30 min at 37°C. Slides were then washed three times for 5 min in PBTW and mounted a final time in VECTASHIELD containing DAPI. The second set of probes were imaged using the exact same parameters as the first.

## SABER-FISH image processing

All image processing was performed in FIJI/ImageJ (*Schindelin et al., 2012*) and plugin source code can be accessed through https://github.com/jayunruh/Jay_Plugins, copy archived at *Unruh, 2023*. Plugins can be used by following the Stowers Fiji update site. Convenience macros that combine these plugin functionalities are included in the supplemental materials.

After acquisition, images were scaled by 0.5 with averaging (i.e. binning). Background subtraction was run on the scaled images using the 'roi average subtract jru v1' plugin. This plugin takes the average intensity signal in a selected region of interest (ROI) or chosen area of background in the image and removes that average across the image. The two series of z-stack images acquired for each set of probes were then aligned and combined through a 'registration_macro.ijm'. In summary, this macro registers the two z-stacks according to DAPI signal from a selected representative z-slice. Alignment is achieved through using an implementation of the TurboReg tool set to a rigid body transformation (*Thévenaz et al., 1998*).

Once the two sets of probes have been aligned, the signals are measured and tracked with the macro 'all_combined_aftersubreg.ijm'. Images were first copied, and then sum projected in z before undergoing a Gaussian blur with sigma value 4. Nuclei are then identified in the DAPI channel by a maximum finding approach with a minimum spot distance of 70 pixels and a threshold of 10% of the maximum intensity (*Varberg et al., 2022*). Those spots provided a locational value for later mapping

of transcript signals in 2D. Individual transcripts are found in 3D in the original combined images with a Gaussian blur with sigma value 1 and a rolling ball background subtraction with a radius of 10 pixels. Because the signal is very punctate and slides can accumulate autofluorescent debris over the course of staining, we next removed the 20 brightest spots in 3D with a spheroid of xy diameter 15 and z diameter 5 to ensure we were evaluating true signal. Spots were found using the same maximum finding approach as above but in 3D with a minimum separation of 12 pixels in xy and 4 slices in z and a threshold at 7% of the maximum intensity in each channel. The positions of those found maxima were then sum projected in z to estimate the number of transcripts in the vicinity of each nuclear maximum (see above).

An image showing the number of transcripts per cell was generated using 'make_nuclear_image.py' macro based on the nuclei and transcript locations identified in the previous step. To generate the polyline kymograph showcasing spatial expression through the tissue, a zoomed-in region of the transcript mapped images was generated around the dorsal neural fold tips. A line of 100 pixel width was drawn starting from the middle of the neuroepithelium toward the dorsal most tip of the neural fold and then ventrally into the underlying mesoderm and migratory NCC population. The polyline kymograph was generated based on this line using the plugin 'polyline kymograph jru v1'.

## RNA in situ hybridization

*Dlc1, Pak3,* and *Sp5* in situ plasmids were previously published (*Dunty et al., 2014*; *Piccand et al., 2014*). RNA in situ hybridization was performed as follows: Embryos were rehydrated through a descending methanol series from 100% methanol to DEPC-PBTW, then washed two times for 5 min in DEPC-PBTW. Embryos were bleached in 6% hydrogen peroxide in the dark for 15 min with rocking and then incubated in 10 µg/ml proteinase K in DEPC-PBTW for 4–5 min at room temperature, without rocking. Embryos were washed with 2 mg/ml glycine in DEPC-PBTW for 5 min, then two times for 5 min in DEPC-PBTW before being refixed in 4% DEPC-PFA + 0.2% glutaraldehyde for 20 min. Embryos were then washed three times for 5 min in DEPC-PBTW, rinsed in prewarmed (68°C) hybridization buffer (50% formamide, 5× SSC pH 4.5, 0.05% EDTA pH 8, 0.2% Tween-20, 0.1% CHAPS, 20 mg/ml Boehringer blocking powder, 1 mg/ml torula RNA, 0.05 mg/ml heparin), and incubated in hybridization buffer with rocking for 1 hr at 68°C. Embryos were then incubated in digoxygenin-labeled ribo-probes (~2 ng/µl) in hybridization buffer overnight at 68°C. Day 2: Embryos were washed two times for 30 min with hybridization buffer at 68°C, followed by prewarmed Solution I (50% formamide, 1× SSC pH 4.5, 0.1% Tween-20), three times for 30 min at 65°C. Embryos were then washed in 50% Solution I/50% MABT (1× MAB – maleic acid, 0.1% Tween-20) for 30 min at room temperature, followed by MABT, three times for 5 min. Embryos were then blocked in MABT + 2% Boehringer blocking powder for 1 hr at room temperature, followed by MABT + 2% Boehringer blocking powder + 20% heat-treated goat serum for 2 hr. Embryos were then incubated overnight in anti-digoxygenin-AP diluted 1:2000 in MABT + 2% Boehringer blocking powder + 20% heat-treated goat serum, at 4°C. The embryos were then rinsed in MABT, washed two times for 15 min in MABT, followed by five to eight further washes in MABT for 1–1.5 hr each, and an overnight wash in MABT at room temperature. The embryos were then washed three times for 10 min in NTMT (100 mM NaCl, 100 mM Tris pH 9.5, 50 mM MgCl$_2$, 0.1% Tween-20), and incubated in NTMT + BCIP/NBT at room temperature in the dark. Color development at room temperature was allowed to continue until the desired darkness of the substrate was achieved. The color reaction was stopped by washing in PBTW, after which the embryos were stored long term in 4% PFA (in PBS)/0.1% glutaraldehyde at 4°C. Stained embryos were imaged and sectioned transversely at 10 µm thickness prior to imaging.

## Aphidicolin treatment

E7.5-E8.0 CD1 embryos were dissected in Tyrode's buffer with their yolk sac intact and incubated in prewarmed culture media for 30 min to 1 hr as previously described (*Muñoz and Trainor, 2019*; *Sakai and Trainor, 2014*; *Sakai and Trainor, 2016*). After 30 min equilibration, 0.5 µg/ml aphidicolin (Sigma-Aldrich, #A0781) was added to the culture media to inhibit S phase of the cell cycle. The same quantity of DMSO was added to the control embryos. Following 12–13 hr of roller culture, EdU was added during the final 15 min. Embryos were then fixed in 4% PFA (in PBS) overnight at 4°C. Fixed embryos were embedded and sectioned at 10 µm and stained using the EdU kit following the manufacturer's instructions. Sox10 and pHH3 fluorescent immunostaining was performed after EdU staining

as described above. TUNEL staining (described above) was performed after fluorescent immunostaining if applicable.

## ShRNA-based lentiviral plasmids and lentivirus production

An shRNA plasmid clone set targeting *Dlc1* was obtained from GeneCopoeia (MSH100727-LVRU6MP). Each set contains three shRNA expression constructs and one scrambled shRNA control. Each shRNA hairpin consists of a 7 base loop and 19–29 base stem optimized for specific gene sequences as detailed by the manufacturer. Glycerol stocks of shRNA-based lentiviral plasmids were cultured in LB buffer with 100 µg/ml of ampicillin (Amresco # 0339). Plasmids were purified using a HiSpeed Plasmid midi kit (QIAGEN #12643).

A total of $4e^6$–$5e^6$ of 293T cells were seeded in one 10 cm plate with 12 ml of media without antibiotics the day before transfection. On the following day, when the cells reached 70–80% confluence, transfection was performed as follows: shRNA-based lentiviral plasmids (7 µg), Pax2 packaging plasmid (7 µg), and VSVG envelop plasmid (1 µg) were mixed with 45 µl of Fugene HD transfection reagent (Promega, cat# E5911) in 1.5 ml of Opti-MEM (Gibco, cat# 31985070). The mixture was incubated at room temperature for 15 min and then added dropwise to the plate of 293T cells. Virus-containing old culture media was harvested after 48 hr and 72 hr upon transfection. We then mixed the virus-containing media and added HEPES to reach 10 mM final concentration. The media was spun down at 500×*g* for 5 min at 4°C to remove cell debris and the supernatant was then collected and filtered through 0.45 µm filter into a falcon tube. We added 1 volume of 4× lentivirus concentrator solution (40% PEG-8000 in 1.2 M NaCl) to 3 volumes of the filtered virus, mixed well and placed the tube on a shaker (60 rpm) at 4°C for overnight. The filtered virus was centrifuged at 1600×*g* for 1 hr at 4°C and the supernatant was carefully removed without disturbing the pellet. The pellet was then thoroughly resuspended with 1 ml of cold PBS by gently pipetting up and down. The solution was transferred to 1.5 ml tube and placed at room temperature for 10 min. We gently pipetted the virus again about 20 times and spun it down in a microcentrifuge at full speed for 3 min to pellet the protein debris. The supernatant was then aliquoted and stored at –80°C for future use.

## Lentivirus injection

E7.5–E8.0 CD1 embryos were dissected in Tyrode's buffer with an intact yolk sac and equilibrated in prewarmed culture media for 30 min as previously described (*Muñoz and Trainor, 2019*; *Sakai and Trainor, 2014*; *Sakai and Trainor, 2016*). Viruses in 5 µl aliquots were thawed and kept on ice, then pipetted onto a piece of parafilm in a small dish with the lid on to prevent evaporation. Using the Eppendorf CellTram Vario system, 1–2 µl of each virus was injected into the amnionic cavity of each embryo (or until the amnion had visibly expanded) after which time, the embryos were returned to roller culture. After 24 hr of incubation, embryos were rinsed in PBS and fixed in 4% PFA (in PBS) at 4°C overnight. Fixed embryos were imaged, embedded, and sectioned at 10 µm thickness before fluorescent immunostaining with Sox10.

## RNA isolation, cDNA preparation, and qRT-PCR

Individual E8.5 mouse embryo head and tail tissues post culturing were collected into 1.5 ml tubes and flash frozen on dry ice with minimal DEPC-Tyrode's buffer remained in the tube. RNA was extracted using the QIAGEN miRNeasy Micro Kit (QIAGEN #217084) with on-column DNase treatment. RNA concentration was determined by Nanodrop. The SuperScript III First-Strand Synthesis System (Invitrogen #18080051) was used to synthesize cDNA for quantitative reverse transcription-PCR (qRT-PCR) with random hexamer primers, and qRT-PCR was performed on an ABI7000 (Thermo QuantStudio 7) using Perfecta SYBR Green (Quantbio #95072-250). Primers are listed in *Supplementary file 2*. No template and no reverse transcription controls were run as negative controls. ΔΔCt method was used to calculate fold change. One-way ANOVA was used for statistical analysis and significance was determined based on $p < 0.05$.

## β-Galactosidase staining

E8.5 *Mef2c-F10N-LacZ* embryos were collected and fixed in 2% formalin, 0.2% glutaraldehyde in 1× PBS for 15–20 min. Embryos were rinsed with PBS and stained according to the manufacturer's

protocol (Millipore #BG-6-B, #BG-7-B, #BG-8-C). Embryos were then fixed again in 4% PFA (in PBS) at 4°C with rocking overnight followed by washing in PBS for whole embryo brightfield imaging.

## Fluorescent imaging

Fluorescently stained section images were captured on an upright Zeiss LSM-700 laser scanning confocal microscope using 405 nm, 488 nm, 555 nm, and 639 nm excitation lasers. Emissions filters used to acquire images were far-red: LP 640 nm, red: BP 505–600 nm, GFP: BP 490–555 nm, DAPI: SP 490 nm. Images were acquired with a Zeiss Fluar 10× objective lens. For each specimen, a z-stack of images was collected and processed as a maximum intensity projection.

All the images used for quantification were acquired with an Orca Flash 4.0 sCMOS 100 fps at full resolution on a Nikon Eclipse Ti2 microscope equipped with a Yokagawa CSU W1 10,000 rpm Spinning Disk Confocal system. The spinning disk confocal is equipped with a quad filter for excitation with 405/488/561/640. Emissions filters used to acquire images were far-red: 669–741 nm, red: 579–631 nm, GFP: 507–543 nm, DAPI: 430–480 nm. A Nikon Plan Apochromat Lambda LWD 40× objective was used to acquire the images with 50–100 ms exposure times.

## Image processing

All analyses of fluorescent intensity were performed using Fiji and custom-written ImageJ Macro and Python notebooks. Prior to the analyses, raw images were processed by subtracting background and were then projected for the max intensity to form single multiple-color images. Individual cells were segmented based on DAPI channel with a pre-trained cellpose model (*Stringer et al., 2021*), and then the mean intensity of the individual channel was measured by FIJI. The segmented cells were classified as either positive or negative based on their intensity in the corresponding channel. For images from the shRNA lentivirus injection experiments, due to the large variety of background signals, we manually labeled positive cells and then trained a cellpose model based on the manually labeled cells. Positive cells were classified based on the trained model, and their coordinates were recorded and saved to ImageJ ROI files for future verification.

For the cell cycle staining analysis, cells in the most dorsal lateral domain of the cranial and trunk neural plate were selected and saved to ImageJ ROI files. In cranial neural plate border cells, the following quantifications were performed: EdU+%=the percentage of EdU positive cells within eYFP positive delaminating premigratory NCC; pHH3+%=the percentage of pHH3 positive cells within eYFP positive delaminating premigratory NCC; EdU+pHH3+%=the percentage of EdU and pHH3 double positive cells within eYFP positive delaminating premigratory NCC; EdU-pHH3-%=the percentage of EdU and pHH3 double negative cells within eYFP positive delaminating premigratory NCC. In trunk neural plate border cells, the following quantifications were performed: EdU+%=the percentage of EdU positive cells within DAPI positive neural plate border cells at the trunk axial level; pHH3+%=the percentage of pHH3 positive cells within DAPI positive trunk neural plate border cells; EdU+pHH3+%=the percentage of EdU and pHH3 double positive cells within DAPI positive trunk neural plate border cells; EdU-pHH3-%=the percentage of EdU and pHH3 double negative cells within DAPI positive trunk neural plate border cells. For the Aphidicolin treatment experiments, neuroepithelial cells and Sox10 positive migratory NCC were selected and saved to ImageJ ROI files. The ratio of Sox10 positive migratory NCC over DAPI positive neural plate/neuroepithelial cells was calculated. For the shRNA lentivirus injection experiments, Sox10 positive migratory NCC were selected and saved to ImageJ ROI files. The number of Sox10 positive cells was quantified.

## Brightfield imaging

Embryos were imaged on a Leica MZ16 microscope equipped with a Nikon DS-Ri1 camera and NIS Elements imaging software. Manual z-stacks were taken and then assembled using Helicon Focus software. Sections from embryos stained by in situ hybridization were imaged on a ZEISS Axio Vert and stitched using ImageJ if needed.

## Acknowledgements

We thank members of the Trainor laboratory for their insights and suggestions on the project. We also acknowledge our amazing animal technician, Melissa Childers, and the Laboratory Animal Services facility at Stowers Institute for Medical Research for animal husbandry and care of the mice used in

this work. The *Dlc1*, *Sp5,* and *Pak3* ISH plasmids were generously provided by Dr. Marian Durkin, Dr. Gerard Gradwohl, and Dr. Terry Yamaguchi, respectively. We also thank Mark Miller, who created the beautiful illustration for the experimental workflow included in *Figure 1*. Funding for this research was provided by the Stowers Institute for Medical Research (PAT) and the National Institute for Dental and Craniofacial Research F31 DE032256 (ELM). The scRNA-seq data was generated through the University of Kansas Medical Center Genomics Core. Funding for the Genomics Core services was provided by the Kansas Intellectual and Developmental Disabilities Research Center (NIH U54 HD 090216), the Molecular Regulation of Cell Development and Differentiation – COBRE (P30 GM122731-03) and the NIH S10 High-End Instrumentation Grant (NIH S10OD021743) at the University of Kansas Medical Center, Kansas City, KS 66160. Original data underlying this manuscript can be accessed from the Stowers Original Data Repository at http://www.stowers.org/research/publications/LIBPB-2442.

## Additional information

### Funding

| Funder | Grant reference number | Author |
| --- | --- | --- |
| Stowers Institute for Medical Research | 1008 | Paul A Trainor |
| National Institute of Dental and Craniofacial Research | | Emma L Moore |
| University of Kansas Medical Center | HD 090216 | Ruonan Zhao Paul A Trainor |
| National Institute of General Medical Sciences | GM122731-03 | Ruonan Zhao Paul A Trainor |
| National Institutes of Health | S10OD021743 | Ruonan Zhao Paul A Trainor |
| National Institute for Dental and Craniofacial Research | F31 DE032256 | Emma L Moore |

The funders had no role in study design, data collection and interpretation, or the decision to submit the work for publication.

### Author contributions

Ruonan Zhao, Conceptualization, Data curation, Formal analysis, Validation, Investigation, Visualization, Methodology, Writing – original draft, Writing – review and editing; Emma L Moore, Conceptualization, Data curation, Formal analysis, Funding acquisition, Validation, Investigation, Visualization, Methodology, Writing – original draft, Writing – review and editing; Madelaine M Gogol, Jay R Unruh, Zulin Yu, Allison R Scott, Data curation, Formal analysis, Investigation, Methodology, Writing – original draft, Writing – review and editing; Yan Wang, Naresh K Rajendran, Formal analysis, Investigation, Methodology, Writing – original draft, Writing – review and editing; Paul A Trainor, Conceptualization, Resources, Supervision, Funding acquisition, Investigation, Methodology, Writing – original draft, Project administration, Writing – review and editing

### Author ORCIDs

Emma L Moore ⓘ http://orcid.org/0000-0003-4116-918X
Madelaine M Gogol ⓘ http://orcid.org/0000-0002-8738-0995
Jay R Unruh ⓘ http://orcid.org/0000-0003-3077-4990
Paul A Trainor ⓘ http://orcid.org/0000-0003-2774-3624

### Ethics

All animal experiments were conducted in accordance with Stowers Institute for Medical Research Institutional Animal Care and Use Committee (IACUC)-approved protocol (IACUC no. 2022-143).

Reviewer #1 (Public Review): https://doi.org/10.7554/eLife.92844.3.sa1

Reviewer #2 (Public Review): https://doi.org/10.7554/eLife.92844.3.sa2
Reviewer #3 (Public Review): https://doi.org/10.7554/eLife.92844.3.sa3
Author response https://doi.org/10.7554/eLife.92844.sa4

## Additional files

### Supplementary files

• Supplementary file 1. Summary table of key neural crest cell (NCC) development related genes and reporters, organized by their expression in the neuroepithelium, neural plate border, and migrating NCC (MNCC), as determined by single-cell RNA sequencing.

• Supplementary file 2. Summary table of forward and reverse primers used for quantitative reverse transcription-PCR (qRT-PCR) of *Dlc1*, *B2M*, *CANX*.

• MDAR checklist

### Data availability

Sequencing data has been deposited in Gene Expression Omnibus (accession no. GSE168351). All original data underlying this manuscript can be freely accessed from the Stowers Original Data Repository at http://www.stowers.org/research/publications/LIBPB-2442.

The following dataset was generated:

| Author(s) | Year | Dataset title | Dataset URL | Database and Identifier |
|---|---|---|---|---|
| Zhao R, Moore EL, Gogol MM, Unruh JR, Yu Z, Scott AR, Wang Y, Rajendran NK, Trainor PA | 2024 | Identification and characterization of intermediate states in mammalian neural crest cell epithelial to mesenchymal transition and delamination | https://www.stowers.org/research/publications/LIBPB-2442 | Stowers Original Data Repository, LIBPB-2442 |

The following previously published dataset was used:

| Author(s) | Year | Dataset title | Dataset URL | Database and Identifier |
|---|---|---|---|---|
| Zhao R, Gogol MM, Trainor PA | 2022 | Single cell RNA-seq of E8.5 mouse embryonic craniofacial tissues | https://www.ncbi.nlm.nih.gov/geo/query/acc.cgi?acc=GSE168351 | NCBI Gene Expression Omnibus, GSE168351 |

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
