## [Editor Report · eLife assessment]

This **fundamental** study reports **compelling** findings that intermediate states exist in epithelial-mesenchymal transition (EMT) during natural development and differentiation of mammalian neural crest cells, similar to recent reports in cancer. The authors determined that there were at least two paths to delamination and migration - one that occurs during S-phase of cell cycle and another during G2/M phase, and that the process of delamination is not restricted to cell fate. Finally, the authors showed that expression of Dlc1 may be used to identify cells in an intermediate state of EMT as well as their spatial location in the mouse embryo. The work will be of interest to developmental biologists, neurobiologists and cancer researchers.

---

## [Referee Report · Reviewer #1 (Public Review)]

Summary:

This describes the molecular identity of the intermediate status of cranial neural crest cells (NCCs) during the initial delamination process. Taking advantage of single-cell RNA seq, the authors identify new populations of cells during EMT characterized by a specific set of gene expressions, including Dlc1. Promigratory cranial NCCs differentiate through different trajectories depending on their cell cycle phases but converge into a common progenitor, then differentiate into mesenchymal cells expressing region-specific genes.

Strengths:

Single-cell RNA seq data convincingly support what the authors claim. This is the first time to identify intermediate states between premigratory and migratory cranial NCCs. Silencing one of the marker genes, Dlc1, reduces the migratory activity of cranial NCCs. These findings deepen our understanding of the mechanism of EMT in general.

Comments on revised version:

Weaknesses:

None after substantial revision.

---

## [Referee Report · Reviewer #2 (Public Review)]

Zhao et al., focus on mechanisms through which cells convert from epithelium to mesenchyme and become migratory. This phenomenon of epithelial-to-mesenchymal transition (EMT) occurs during both embryonic development and cancer progression. During cancer progression, EMT seemingly includes cells at intermediate states as defined by the combinatorial expression of epithelial and mesenchymal markers. But the importance of these markers and the role of these intermediate states remains unclear. Moreover, whether EMT during development also involves equivalent intermediate cell states is not known. To address this gap in knowledge, the authors devise a strategy to identify and characterize changes that an embryonic population of cells called the cranial neural crest undergo as they delaminate from the neuroepithelium and become a highly migratory population of mesenchymal cells that ultimately give rise to a broad range of derivatives.

To isolate and study the neural crest, the authors use embryos collected at E8.5 from two transgenic mouse lines. Wnt1-Cre;RosaeYFP labels Wnt1-positive neuroepithelial cells in the dorsolateral neural plate, which includes pre-migratory neural crest that reside in the dorsal neuroectoderm and neural plate border before induction (as well as some other lineages). Mef2c-F10N-LacZ leverages a neural crest cell-specific enhancer of Mef2c to control LacZ expression in predominantly migratory neural crest. This dual genetic approach that allows the authors to distinguish and compare pre-migratory and migratory neural crest cells is a strength of the work.

To assay for the differential expression of genes involved in the EMT and migration of cranial neural crest, the authors perform single cell RNA sequencing (scRNA-seq) using current methods. A strength is a large sample size per mouse line, and relatively high numbers of single cells analyzed. The authors identify six major cell/tissue types present in mouse E8.5 cranial tissues using known markers, which they then segregate into a cranial neural crest cluster using a well-reasoned bioinformatic strategy. The cranial neural crest cluster contains pre-migratory and migratory cells that they partition further into five subclusters and then characterize using the differential expression and combinatorial patterns of neural crest specifier genes, markers of pre-migratory neural crest, markers of early versus late migratory neural crest, markers of undifferentiated versus differentiated neural crest, tissue-specific markers, and region-specific markers. One weakness is that there is little attempt to map potential novel genes and/or pathways that also distinguish these clusters.

The authors then go on to subdivide the five cranial neural crest subclusters into almost two dozen smaller subclusters, again using the combinatorial expression of known markers (e.g., neural crest genes, cell junction genes, and cell cycle genes). A weakness is that the marker analysis and accompanying interpretation of the results relies heavily on the purported roles of different genes as described in the published work of others, which potentially introduces some untested assumptions and a bit of hand-waving into the study. Moreover, the limited correlation between mRNA and protein abundance for cell cycle markers is well documented in the literature but the authors rely heavily on gene expression to determine cell cycle status. Even though the authors add a compelling Edu/pHH3 double-labeling experiment and cell cycle inhibition studies, the work would be strengthened by including some analysis of protein expression to see if the cell cycle correlations hold up. Nonetheless, the subcluster and cell cycle analyses lead the authors to conclude that there are a series of intermediate cell states between neural crest EMT and delamination, and that cell cycle regulation is a defining feature and necessary component of those states. These novel findings are generally well supported by the data.

To test if there are spatiotemporal differences in the localization of neural crest cells during EMT in vivo, the authors apply a cutting-edge technique called signal amplification by exchange reaction for multiplexed fluorescent in situ hybridization (SABER-FISH), which they validate using standard in situ hybridization. The authors select specific marker genes that seem justified based on their scRNA-seq dataset, and they generate a series of convincing images and quantitative data that add valuable depth to the story.

As a functional test of their hypothesis that one of the genes indicative of an EMT intermediate stage (i.e., Dlc1) is essential for neural crest migration, the authors use a lentivirus-mediated knockdown strategy. A strength is that the authors include appropriate scramble and cell death controls as part of their experimental design.

The authors use Sox10 as a marker to count neural crest cells, but Sox10 may only label a subset of neural crest cells and thus some unaffected lineages may not have been counted. Although the data are persuasive, a second marker for counting neural crest cells following knockdown would make the analysis more robust.

Overall, this is a first-rate study with many more strengths than weaknesses. The authors generate high quality data, and their interpretations are reasonable and balanced. Another strength is the writing, which is clear and well organized, and the figures (including supplemental), which are excellent and provide unambiguous visualization of some very complex data sets. The methods are state-of the art and are effectively executed, and they will be useful to the broader cell and developmental biology community. The work contains well-substantiated findings and supports the conclusion that EMT is a highly dynamic, multi-step process, which was previously thought to be more-or-less binary. Such findings will alter the way the field thinks about EMT in neural crest and the work will likely serve as an important example alongside cancer metastasis.

---

## [Referee Report · Reviewer #3 (Public Review)]

Summary:

Zhao et al. address the question of whether intermediate states of the epithelial-to-mesenchymal transition (EMT) exist in a natural developmental context as well as in cancer cells. This is important not only for our understanding of these developmental systems but also for their development as resources for new anti-cancer approaches. Guided by single-cell RNA sequencing analysis of delaminating mouse cranial neural crest cells, they identify two distinct populations with transcriptional signatures intermediate between neuroepithelial progenitors and migrating crest. Both clusters are also spatially intermediate and are actively cycling, with one in S-phase and one in G2/M. They show that blocking progression through S phase prior to the onset of delamination and knockdown of intermediate state marker Dlc1 both reduce the number of migratory cells that have completed EMT. Overall, the work provides a modern take and new insights into the classical developmental process of neural crest delamination.

Strengths:

• Deep analysis of the scRNAseq dataset revealed previously unappreciated cell populations intermediate between premigratory and migratory crest.

• The observation that delaminating/intermediate neural crest cells appear to be in S or G2/M phase is interesting and worth reporting, though the ultimate significance remains unclear, given that they do not make distinct derivatives depending on their cycle state.

• The authors employ new methods for multiplex spatial imaging to more accurately define their populations of interest and their relative positions.

• The authors present evidence that intermediate state gene Dlc1 (a Rho GAP) is not just a marker but functionally required for neural crest delamination in mouse, as previously shown in chicken.

Weaknesses:

• Similar experiments involving blockade of cell cycle progression and Dlc1 dose manipulation were previously performed in chick models, as noted in the discussion. The newly-defined intermediate states give added context to the results, but they are not entirely novel.

---

## [Author Response]

The following is the authors’ response to the original reviews.

**Reviewer #1 (Recommendations For The Authors):**
(1) More explanation/description of Fig 3C and 3D would be helpful for readers, including the color code of 3D and black lines shown in both panels.

We have added more description to the legend of Figure 3, and we have used the same color code as in Figure 2, which we now specifically note in the figure legend as well.

(2) Differences between cranial and trunk NCC could be experimentally shown or discussed. Fig 4C shows some differences between these two populations, but in situ, results using Dlc1/Sp5/Pak3 probes in the trunk region may be informative, like Fig 5 supplement 2 for cranial NCCs.

This is an important point. The focus of our study was on cranial neural crest cells, and the single cell sequencing data is therefore truly reflective of only cranial neural crest cells. We have not functionally tested for the roles of Dlc1/Sp5/Pak3 in trunk neural crest cells, however, based on the expression and loss-of-function phenotypes of Sp5 or Pak3 knockout mice, we predict they individually may not play a significant role. It remains plausible that Dlc1 could play an important role in the delamination of trunk neural crest cells, but we have not tested that definitively. Nonetheless, Sabbir et al 2010 showed in a gene trap mouse mutant that Dlc1 is expressed in trunk neural crest cells. Regarding the similarities and differences between cranial and trunk neural crest cells as noted by the reviewer with respect to Figure 4, it’s important to recognize the temporal differences illustrated in Figure 4. Neural crest cell delamination proceeds in a progressive wave from anterior to posterior, but also that the analysis was designed to quantify cell cycle status before and during neural crest cell delamination. We have compared cranial and trunk neural crest cells in more detail in the discussion and also speculate what might happen in the trunk based on what we know from other species.

(3) Discussion can be added about the potential functions of Dlc1 for NCC migration and/or differentiation based on available info from KO mice.

We have added specific details regarding the published Dlc1 knockout mouse phenotype to the discussion, particularly with respect to the craniofacial anomalies which included frontonasal prominence and pharyngeal arch hyperplasia, and defects in neural tube closure and heart development. Although the study didn’t investigate the mechanisms underpinning the Dlc1 knockout phenotype, the craniofacial morphological anomalies would be consistent with a deficit in neural crest cell delamination reducing the number of migrating neural crest cells, as we observed in our Dlc1 knockdown experiments.

**Reviewer #2 (Recommendations For The Authors):**
The authors used the (Tg(Wnt1-cre)11Rth Tg(Wnt1-GAL4)11Rth/J) line but work from the Bush lab (see Lewis et al., 2013) has demonstrated fully penetrant abnormal phenotypes that affect the midbrain neuroepithelium, increased CyclinD1 expression and overt cell proliferation as measured by BrdU incorporation. The authors should explain why they used this mouse line instead of the Wnt1-Cre2 mice (129S4-Tg(Wnt1-cre)1Sor/J) in the Jackson Laboratory (which lacks the phenotypic effects of the original Wnt1-Cre line), or a "Cre-only" control, or at a minimum explain the steps they took to ensure there were no confounding effects on their study, especially since cell proliferation was a major outcome measure.

This is an important point, and we thank the reviewer for raising it. Yes, it has been reported that the original Wnt1Cre mice exhibit a midbrain phenotype (Ace et al. 2013). However, it has also been noted that Wnt1Cre2 can exhibit recombination in the male germline leading to ubiquitous recombination (Dinsmore et al., 2022). Therefore, to avoid any potential for bias, we used an equal number of cells derived from the Wnt1 and F10N transgenic line embryos in our scRNA-seq, and this included multiple non-Cre embryos. Our scRNA-seq analysis was therefore not dependent upon Wnt1-Cre, but also because we used whole heads not fluorescence sorted cells. However, Wnt1-Cre lineage tracing was advantageous from a computational perspective to help define cells that were premigratory and migratory in concert with Mef2c-lacZ ¬based on their expression of YFP, LacZ or both. We note these specifics more clearly in the methods.

The Results section (line 122) states that scRNA-seq was performed on dissociated cranial tissues but the Methods section (lines 583-584) implies that whole E8.5 mouse embryos were dissociated. Which was dissociated, whole embryos or just cranial tissues? Obviously, the latter would be a better strategy to enrich for cranial neural crest, but the authors also examine the trunk neural crest. This should be clarified in the text.

We apologize that some of the details regarding the tissue isolation were confusing and we have clarified this in the methods and the text. For the record, after isolating E8.5 embryos, we then dissected the head from those embryos, and performed scRNA-seq on dissociated cranial tissues. As the reviewer correctly noted, this approach strategically enriches for cranial neural crest cells.

The authors do not justify why they chose a knockdown strategy, which has its limitations including its systemic injection into the amniotic cavity, its likely global and more variable effects, and its need to be conducted in culture. Why the authors did not instead use a Wnt1-Cre-mediated deletion of Dlc1, which would have been "cleaner" and more specific to the neural crest, is not clear (maybe so they could specifically target different Dcl1 isoforms?). Also, the authors use Sox10 as a marker to count neural crest cells, but Sox10 may only label a subset of neural crest cells and thus some unaffected lineages may not have been counted. The authors should mention what is known about the regulation of Dcl1 by Sox10 in the neural crest. Although the data are persuasive, a second marker for counting neural crest cells following knockdown would make the analysis more robust. Can the authors explain why they did not simply use the Mef2c-F10N-LacZ line and count LacZ-positive cells (if fluorescence signal was required for the quantification workflow, then could they have used an anti-beta Galactosidase antibody to label cells)?

We thank the reviewer for raising these important considerations. It has previously been noted that although Wnt1-Cre is the gold standard for conditional deletion analyses in neural crest cell development, especially migration and differentiation, it is not a good tool for functional studies of the specification and delamination of neural crest cells due to the timing of Wnt1 expression and Cre activation and excision (see Barriga et al., 2015). Therefore, we chose a knockdown strategy instead, and also because it allows us to more rapidly evaluate gene function. We agree that there are limitations to the approach with respect to variability, however, this is outweighed by the ability to repeatedly perform the knockdown at multiple and more relevant temporal stages such as E7.5 (which is prior to the onset of Wnt1-Cre activity), as well as target different isoforms, and also treat large numbers of embryos for quantitative analyses. The advantage of using Sox10 as a marker for counting neural crest cells is that at the time of analysis, cranial neural crest cells are still migrating towards the frontonasal prominences and pharyngeal arches, and the overwhelming majority of these cells are Sox10 positive. Moreover, we can therefore assay every Dlc1 knockdown embryo for Sox10 expression and count the number of migrating neural crest cells. The limitation of using the Mef2c-F10N-LacZ line is that this transgenic line is maintained as a heterozygote, and thus only half the embryos in a litter could reasonably be expected to be lacZ+. But combining Sox10 and Mef2c-F10N-LacZ fluorescent immunostaining for similar analyses in the future is a great idea.

**Reviewer #3 (Recommendations For The Authors):**
The putative intermediate cells differentially express mRNAs for genes involved in cell adhesion, polarity, and protrusion relative to bona fide premigratory cells (Fig. 2E). This is persuasive evidence, but only differentially expressed genes are shown. Discussing those markers that have not yet changed, e.g. Cdh1 or Zo1 (?), would be instructive and help to clarify the order of events.

We thank the author for this suggestion and we have provided more detail about adherens junction and tight junctions. Cdh1 is not expressed, and although Myh9 and Myh10 are expressed, we did not detect any significant changes. ZO1 is a tight junction protein encoded by the gene Tjp1, which along with other tight junctions protein encoding genes, is downregulated in intermediate NCCs as shown in the Figure 2E.

It is unclear whether the two putative intermediate state clusters differ other than their stage of the cell cycle. Based on the trajectory analysis in Fig. 3C-D, the authors state that these two populations form simultaneously and independently but then merge into a single population. However, without further differential expression, it seems more plausible that they represent a single population that is temporarily bifurcated due to cell cycle asynchrony.

We have addressed the cell cycle question in the discussion by noting that while it is possible the transition states represent a single population that is temporarily bifurcated due to cell cycle asynchrony, if this were true, then we should expect S phase inhibition to eliminate both transition state groups. Instead, our trajectory analyses suggest that the transition states are initially independent, and furthermore, S phase inhibition did not affect delamination of the other population of neural crest cells.

The authors do not present an in-depth comparison of these neural crest intermediate states to previously reported cancer intermediate states. This analysis would reveal how similar the signatures are and thus how extrapolatable these and future findings in delaminating neural crest are to different types of cancer.

We have also added more detail to the discussion to address the potential for similarities and differences in neural crest intermediate states compared to previously reported cancer intermediate states. The challenge, however, is that none of the cancer intermediate states have been characterized at a molecular level. Nonetheless, with the limited molecular markers available, we have not identified any similarities so far, but our datasets are now available for comparison with future cancer EMP datasets.

The reduction in SOX10+ cells may be in part or wholly attributable to inhibition of proliferation AFTER delamination. Showing that there are premigratory NCCs in G2/M at ~E8.0 would bolster the argument that this population is present from the earliest stages.

The presence of premigratory neural crest cells in G2/M is shown by the scRNA-seq data and cell cycle staining data in the neural plate border.

Lines 248-249: The pseudo-time analysis in Fig 3C/D does indicate that the two most mature cell clusters (pharyngeal arch and frontonasal mesenchyme) may arise from common or similar migratory progenitors. However, given the decades of controversy about fate restriction of neural crest cells, the statement that "EMT intermediate NCC and their immediate lineages are not fate restricted to any specific cranial NCC derivative at this timepoint" should be toned down so as to not give the impression that they have identified common progenitors of ectomesenchyme and neuro/glial/pigment derivatives.

We appreciate this comment, because as the reviewer noted, there has been considerable literature and debate about the fate restriction and plasticity of neural crest cells, and indeed we did not intend to imply we have identified common progenitors of ectomesenchyme and neuro/glial/pigment derivatives. That can only be truly functionally demonstrated by clonal lineage tracing analyses. Rather, we interpret our pseudo-time analyses to indicate that irrespective of cell cycle status at the time of delamination, these two populations come together with equivalent mesenchymal and migratory properties, but in the absence of fate determination in the collective of cells. This does not mean that individual cells are common progenitors of both ectomesenchyme and neuro/glial/pigment derivatives. The nuance is important, and we address this more carefully in the text.

Lines 320-321: "...this overlap in expression was notably not observed in older embryos in areas where EMT had concluded". It is unclear whether the markers no longer overlap in older embryos (i.e. segregate to distinct populations) or are simply no longer expressed.

The data in Figure 5 demonstrates the dynamic and overlapping expression of Dlc1, Sp5 and Pak3 in the different clusters of cells as they transition from being neuroepithelial to mesenchymal. In contrast to Sp5 and Pak3, Dlc1 is not expressed by premigratory neural crest cells but is expressed at high levels in all EMT intermediate stage neural crest cells. Later as Dlc1 continues to be expressed in migrating neural crest cells, Pak3 and Sp5 are downregulated. But the absence of overlapping expression in the dorsolateral neural plate at the conclusion of EMT coincides with their downregulation in that territory.

In the final results section on Dlc1, the previously published mutant mouse lines are referenced as having "craniofacial malformation phenotypes". The lack of detail given on what those malformations are (assuming descriptions are available) makes the argument that they may be related to insufficient delamination less persuasive. The degree of knockdown correlates so well with the percentage reduction in migratory neural crest (Fig. 6) that one would imagine a null mutant to have a very severe phenotype.

The inference from the reviewer is correct and indeed Dlc1 null mutant mice do have a severe phenotype. We have added more specific details regarding the craniofacial and other phenotypes of the Dlc1 mutant mice to the discussion. Of note the frontonasal prominences and the pharyngeal arches are hypoplastic in E10.5 Dlc1 mutant embryos, which would be consistent with a neural crest cell deficit. Although a deficit in neural crest cells can be caused my multiple distinct mechanisms, our Dlc1 knockdown analyses suggest that the phenotype is due to an effect on neural crest cell delamination which diminishes the number of migrating neural crest cells.

Use the same y-axis for Fig. 4C/D

This has been corrected.

Fig. 6C: Please note in the panel which gene is being measured by qPCR

This has been corrected to denoted Dlc1.

Lines 108-117: More concise language would be appropriate here.

As requested, we were more succinct in our language and have shortened this section.

The SABER-FISH images are very dim. I realize the importance of not saturating the pixels, but the colors are difficult to make out.

We thank the reviewer for pointing this out and have endeavored to make the SABER-FISH images brighter and easier to see.